



# The influence of a vegetated bar on channel-bend flow dynamics

Sharon Bywater-Reyes[1,2], Rebecca M. Diehl[1], Andrew C. Wilcox[1]

[1]Department of Geosciences, University of Montana, 32 Campus Drive #1296, Missoula, Montana, 59812-1296, USA
[2]Department of Earth and Atmospheric Sciences, 501 20 St., Greeley, Colorado, 80639, USA

*Correspondence to*: Sharon Bywater-Reyes (sharon.bywaterreyes@unco.edu)

The authors declare that they have no conflict of interest.

**Abstract.** Alternating bars influence hydraulics, morphodynamics, and channel geometry in alluvial rivers. Recruitment of pioneer woody riparian vegetation is tightly coupled with bar building, yet the influence of vegetation on changing bend hydraulics and forces has been unresolved. We use a two-dimensional hydraulic model that accounts for vegetation drag to

test the sensitivity of channel-bend hydraulics to riparian vegetation for a gravel-bed river with bars. A calibrated model for the Bitterroot River, Montana (United States) run with and without varied vegetation parameters on a bar shows vegetation slows flow upstream of the bar, steers the high-velocity core of flow toward the cutbank, and creates a large gradient in cross-stream velocity. Results are consistent with a feedback in channels with vegetated bars whereby vegetation steers flow towards the opposite bank, increasing bank erosion at the mid- and downstream end of the bend, and simultaneously increasing rates

of bar accretion through reduction in velocity. Collectively, these patterns of morphodynamics influence topographic steering and channel migration rates.

## 1 Introduction

Channel-bend morphodynamics along meandering rivers influence channel morphology, river migration rates, channel-floodplain connectivity, and aquatic habitat. River bars, fundamental to channel bend morphology (Blondeaux and Seminara,

1985; Ikeda et al., 1981), steer flow and induce convective accelerations (Dietrich and Smith, 1983) that influence boundary shear stress (Dietrich and Whiting, 1989) and sediment transport fields (Dietrich and Smith, 1983; Legleiter et al., 2011; Nelson and Smith, 1989). Erosion of banks and deposition of bars drives the process of channel migration.

Channel-bend and bar dynamics can be tightly coupled with the recruitment and succession of riparian vegetation on river bars (Amlin and Rood, 2002; Eke et al., 2014; Karrenberg et al., 2002; Nicholas et al., 2013; Rood et al., 1998). Plants

change local hydraulics (Nepf, 2012; Rominger et al., 2010) and sediment transport conditions (Curran and Hession, 2013; Manners et al., 2015; Yager and Schmeeckle, 2013), resulting in strong feedbacks between the recruitment and growth of woody riparian vegetation and bar building (Bendix and Hupp, 2000; Dean and Schmidt, 2011) that can impact the hydraulics and morphology of rivers at multiple scales (Curran and Hession, 2013; Osterkamp et al., 2012). Plant traits such as height, frontal area, and stem flexibility vary with elevation above the baseflow channel and influence both the susceptibility of plants



to uprooting during floods and their morphodynamic effects (Bywater-Reyes et al., 2015, 2017; Diehl et al., 2017a; Kui et al., 2014). Vegetation's impact on altering the velocity (flow steering) is poorly understood despite advances in understanding the reciprocal interactions between riparian vegetation recruitment and river processes (Corenblit et al., 2007; Gurnell, 2014; Osterkamp and Hupp, 2010; Schnauder and Moggridge, 2009).

5        In computational modelling of flow and sediment transport, vegetation's effect on river morphodynamics can be simulated with a range of approaches. Reduced complexity models that approximate the physics of flow and sediment transport have successfully reproduced many of the features observed in channels influenced by vegetation, such as the development of a single-thread channel (e.g., Murray and Paola, 2003). Two-dimensional models that use shallow-water equations and, in some cases, sediment transport relations, provide an alternative that may be less dependent on initial conditions and more capable of representing the physics on vegetation-flow interactions (Boothroyd et al., 2016, 2017; Marjoribanks et al., 2017; Nelson et al., 2016; Nicholas et al., 2013; Pasternack, 2011; Tonina and Jorde, 2013). Investigations of channel-bend dynamics influenced by vegetation using two-dimensional models often represent vegetation by increasing bed roughness (see Green, 2005 and Camporeale et al., 2013 for comprehensive reviews). Nicholas et al. (2013) simulated bar and island evolution in large anabranching rivers using a morphodynamic model of sediment transport, bank erosion, and floodplain development on a multi-century timescale where vegetation was modelled using a Chezy roughness coefficient. Asahi et al. (2013) and Eke et al. (2014) modelled river bend erosional and depositional processes that included a bank-stability model and deposition dictated by an assumed vegetation encroachment rule. Bertoldi and Siviglia (2014) used a morphodynamic model coupled with a vegetation biomass model, which accounted for species variations in nutrient and water needs to simulate the coevolution of vegetation and bars in gravel-bed rivers. Vegetation was modelled as increased bed roughness via the Strickler-Manning relation that varied linearly with biomass. Their model showed two scenarios: one where flooding completely removed vegetation, and one where vegetation survived floods, resulting in vegetated bars. These two alternative stable states (bare versus vegetated bars) have been found experimentally as well (Wang et al., 2016).

When vegetation drag is dominant over bed friction, using conventional resistance equations (e.g., Manning's n; roughness) to model vegetation's effect on the flow introduces error. Increasing the roughness within vegetated zones increases the modelled shear stress and therefore artificially inflates the sediment transport capacity at the local scale (e.g., vegetation patch or bar), although at the reach scale the results may be appropriate (Baptist et al., 2005; James et al., 2004). Alternative approaches include accounting for vegetation drag explicitly as cylinders (e.g., Baptist et al., 2007; Vargas-Luna et al., 2015a) or accounting for foliage, streamlining, and the altered drag that occurs as a result (e.g., Boothroyd et al., 2015, 2017; Jalonen et al., 2013; Västilä and Järvelä, 2014). Vargas-Luna et al. (2015a) showed through coupling of numerical modeling and experimental work that representing vegetation as cylinders is most appropriate for dense vegetation. Iwasaki et al. (2015) used a two-dimensional model that accounted for vegetation drag to explain morphological change of the Otofuke River, Japan, caused by a large flood event in 2011 that produced substantial channel widening and vegetation-influenced bar building. They found that vegetation allowed bar-induced meandering to maintain moderate sinuosity, whereas in the absence of vegetation, river planform would switch from single-thread to braided.



Here we evaluate the influence of bar vegetation on channel-bend hydraulics using a two-dimensional hydrodynamic model capable of spatially defining vegetation drag. We model a range of vegetation densities and plant morphologies representing different stages of pioneer woody vegetation growth on a single channel bar. We vary discharge in the model to represent the stage-dependent effects of vegetation on hydraulics, as well as different flood stages that may be important for

the recruitment of plants and the erosion or deposition of sediment within the channel bend. We predict that the presence of woody vegetation affects bar and meander dynamics by steering flow, thereby influencing the morphodynamic evolution of vegetated channels. Our objectives are to 1). Determine which vegetation morphology and flow conditions result in the greatest changes to channel-bend hydraulics; and 2). Infer how these changes in hydraulics would impact channel-bend morphodynamics and evolution. The insights derived from our analyses are relevant to understanding ecogeomorphic

feedbacks between riparian ecosystems and physical processes in meandering rivers, to understanding how such feedbacks are mediated by plant traits and flow conditions, and to managing for different riparian plant species along river corridors.

## 2 Methods

### 2.1 Study area

To meet our objectives, we model a bar-bend sequence on the Bitterroot River, southwest Montana, United States (Fig. 1).

Our field site has a pool-riffle morphology and a wandering pattern, with channel bends, alternate bars, and woody vegetation on bars and floodplains. The study reach is located on a private reserve (MPG Ranch) with minimal disturbance to the channel and floodplain, and flow and sediment supply are relatively unregulated because few upstream dams are present. Annual mean discharge is 68 m$^3$ s$^{-1}$, bankfull Shields number is 0.01, and median grain size is 23 mm. Woody bar vegetation is composed of sand bar willow (*Salix* exigua), and cottonwood (*Populus* trichocarpa) seedlings, saplings, and young trees (Fig. 2a, 2c).

Ponderosa pine (*Pinus* ponderosa), gray alder (*Alnus* incana), and black cottonwood (*Populus* trichocarpa) comprise mature floodplain forest species.





**Figure 1. Bitterroot River, Montana showing model domain, showing location of ADCP velocity measurement cross sections, downstream boundary, tree crowns mapped from airborne LiDAR, the location of the vegetated bar, and the three bars shown in Fig. 9. Inset map shows location in northwestern USA.**



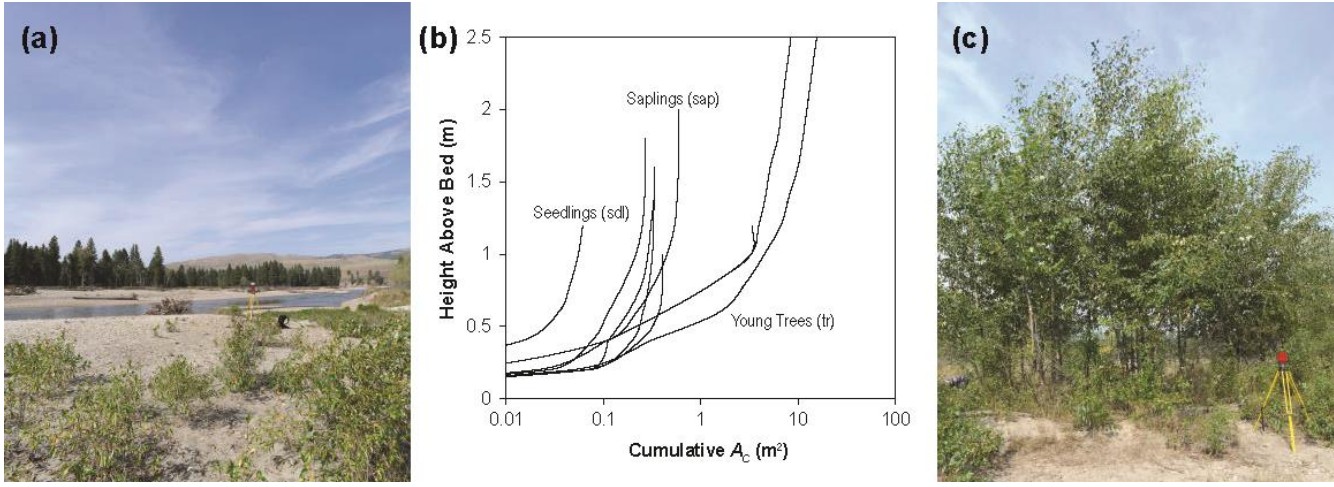

**Figure 2. Modelled vegetated bar (a) on the Bitterroot River, showing sparse *Populus* seedlings and saplings. Cumulative $A_c$ of *Populus* varies with height above the bed, and the age and size of the individual (b); the greatest cumulative $A_c$ is reached for young trees (c). The average profile for seedlings (sdl), saplings (sap), and young trees (tr) was used to assign an $A_s$ value based on flow depth for each run. Photo credit: Sarah Doelger.**



## 2.2 Flow model

To characterize the influence of a vegetated bar on channel-bend hydraulics, we used FaSTMECH (version 2.3.2), a hydrostatic, quasi-steady flow model contained within iRIC (Nelson et al., 2016; http://i-ric.org/en/index.html). FaSTMECH solves the depth- and Reynolds-averaged momentum equations in the streamwise ($s$) and cross-stream ($n$) directions, in a

channel-fitted curvilinear coordinate system, using a finite-difference solution (Nelson et al., 2003, 2016). Bed stress closure is achieved through a drag coefficient scheme ($C_d$), where boundary shear stress ($\tau$) in the streamwise ($s$) and stream-normal ($n$) directions are estimated as:

$$\tau_s = \rho C_d u \sqrt{(u^2 + v^2)} \quad (1)$$

$$\tau_n = \rho C_d v \sqrt{(u^2 + v^2)} \quad (2)$$

where $\rho$ is density of water, and $u$ and $v$ are the velocity components in the streamwise and stream-normal directions, respectively. By convention, values of $u$ and $\tau_s$ are positive downstream, and $v$ and $\tau_n$ positive toward the left bank.

We created the flow model domain in FaSTMECH by characterizing the topography and flow boundary conditions (discharge and water surface elevation at the downstream boundary) of a study reach on the Bitterroot River, Montana (Fig. 1). Topography was surveyed with a combination of airborne LiDAR and RTK GPS (see Supplement for more detail). To

develop a stage-discharge relationship, we linked transducer stage measurements at the downstream end of the study reach to discharge derived from USGS 12344000 Bitterroot River near Darby MT, corrected by contributing area for our field site. Discharge was measured at the field site and compared to the adjusted USGS 12344000 value and found to agree within 10 % (Table 1).

**Table 1. Calibration flows, showing the channel drag ($C_d$) and lateral eddy viscosity ($LEV$), and the root mean square error (RMSE), water surface elevation (WSE), and depth-averaged velocity ($\bar{U}$).**

| Discharge[a] (m³ s⁻¹) | $C_d$ | LEV | RMSE-WSE (m) | RMSE-$\bar{U}$[b] (m s⁻¹) | Vegetation Model |
|---|---|---|---|---|---|
| 48 | 0.003 | 0.04 | 0.11 | NA | Off |
| 62 | 0.003 | 0.004 | 0.11 | 0.29 | Off |
| 62[c] | 0.003 | 0.04 | 0.11 | 0.24[d] | Off |
| 62 | 0.003 | 0.4 | 0.13 | 0.36 | Off |
| 90 | 0.003 | 0.04 | 0.17 | NA | Off |
| 453[d] | 0.003 | 0.04 | 0.16 | NA | Off |
| 453[d] | 0.003 | 0.04 | 0.18 | NA | On |

[a]Corrected by contributing area from USGS 12344000

[b]Law-of-the-wall derived

[c]Discharge measured at site was within 10% of contributing-area-corrected discharge

[d]Q$_2$ flow





We calibrated channel characteristics (bed roughness specified as $C_d$ and lateral eddy viscosity, *LEV*) and considered them fixed after calibration (Table 1). We used a constant $C_d$, an approach that has been shown elsewhere to perform comparably to variable roughness in FaSTMECH (e.g., Segura and Pitlick, 2015). We set $C_d$ to minimize the root mean square

5   error (RMSE) of modelled water surface elevation (WSE) versus WSE measured in the field from 2011–2015, over a range of calibration flows. This process resulted in a $C_d$ of 0.003 and lowest RMSE's for WSE from 0.11 to 0.18 m for the lowest and highest calibration flows, respectively (Table 1). Similarly, we set *LEV* at 0.04, a value that minimized RMSE of depth-averaged velocity ($\bar{U}$=0.24 m s$^{-1}$; Table 1) between modelled values and those measured at four cross sections (Fig. 1; see Supplement for more detail). The RMSE ranges obtained through calibration are consistent with values reported in other

10   studies that have used FaSTMECH (e.g., Legleiter et al., 2011; Mueller and Pitlick, 2014; Segura and Pitlick, 2015), providing confidence in model performance. Relaxation coefficients were set to 0.5, 0.3, and 0.1 for ERelax, URelax, and ARelax, respectively, through trial and error. Convergence was found after 5000 iterations (mean error discharge < 2 %).

To address the stage-dependent nature of the impact of a vegetated bar in altering bend hydraulics, we modelled flows with magnitudes corresponding to flows with return periods of 2 ($Q_2$; 453 m$^3$ s$^{-1}$), 10 ($Q_{10}$; 650 m$^3$ s$^{-1}$), 20 ($Q_{20}$; 715 m$^3$ s$^{-1}$) and

15   100 ($Q_{100}$; 800 m$^3$ s$^{-1}$) years. Because we were unable to maintain a curvilinear, channel-fitted grid (nodes overlapped) but were interested in quantifying hydraulics with respect to a channel bend where such a grid is more appropriate, we converted Cartesian coordinate velocity ($U_x$, $U_y$) to streamwise and stream-normal values (Fig. 3).





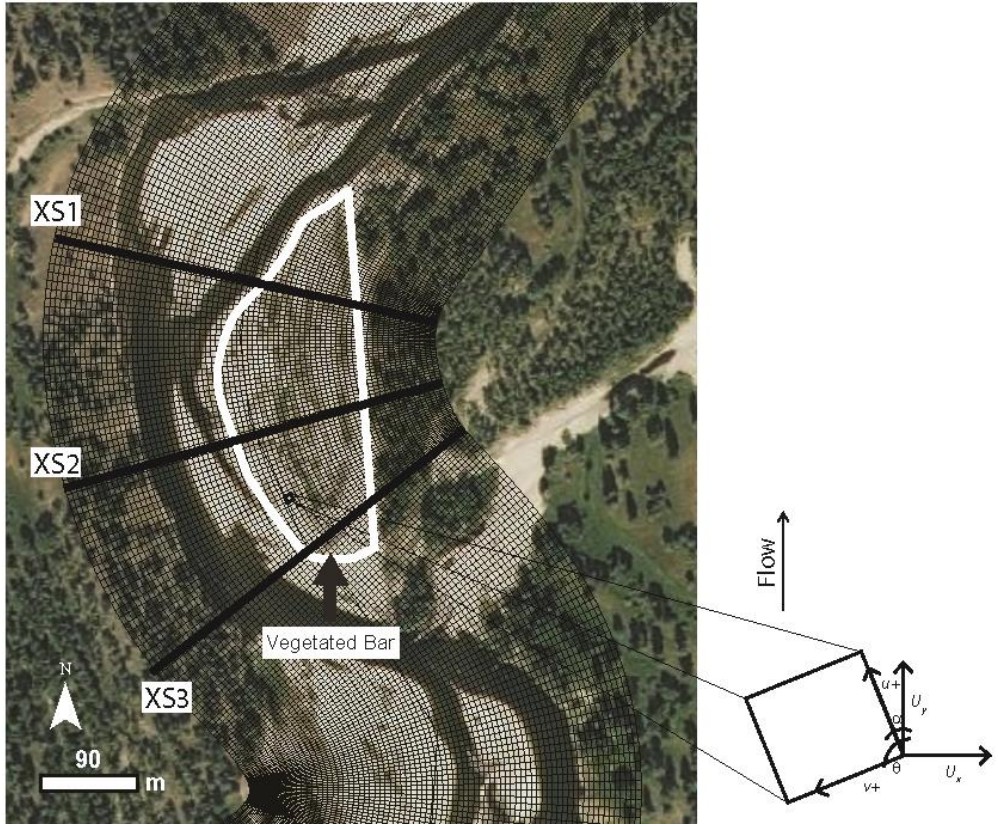

**Figure 3. Region around the vegetated bar, showing cross section (XS) locations and the conventions of the curvilinear grid to which model output was converted.**

5 **2.3 Modelling vegetation's impact on channel-bend hydraulics**

In FaSTMECH we accounted for vegetation form drag ($F_D$) using the following drag equation for rigid vegetation:

$$F_D = \tfrac{1}{2}\rho C_{d,v} A_c U_c^2 \quad (3)$$

where $C_{d,v}$ is vegetation drag coefficient, $A_c$ is projected vertical frontal area of vegetation (Nepf, 1999; Vargas-Luna et al., 2015, 2016), and $U_c$ is the approach velocity. For $U_c$ we substituted cross-sectional mean velocity, $U_m$ (after Jalonen et al.,

10 2013). The vegetation drag coefficient ($C_{d,v}$) was assigned a value of one, a first-order approximation also used by others (Boothroyd et al., 2016; Nepf et al., 2013; Vargas-Luna et al., 2016). We modelled vegetation as cylinders by assuming the cylindrical stem frontal area is equal to $A_c$, specifying vegetation parameters by polygon with an associated stem density (#stems m$^{-2}$) and height (m; allows for partitioning of $A_c$ by flow depth). A logarithmic velocity profile was assumed regardless of whether vegetation was emergent or submerged.



We focused our analyses on a bar (Fig. 1) that supports woody riparian vegetation (*Populus* seedlings, saplings, and young trees). In our model simulations, we varied vegetation density (#stems m$^{-2}$) and $A_c$ (m$^2$ per plant) on the vegetated bar for each of the four flows, and we compared model output to a no-vegetation (no veg) scenario. We considered two vegetation density cases: sparse (*sps*) and dense (*dns*). Our sparse case was based on the average density (0.02 stems m$^{-2}$) obtained from the airborne LiDAR (see Supplement for more detail). Our dense case (20 stems m$^{-2}$) was based on the average from random vegetation density plots measured on the bar, which ranged from <1 stem m$^{-2}$ to 227 stems m$^{-2}$. For $A_c$, we used ground-based LiDAR to capture vegetation structure (Antonarakis et al., 2010; Bywater-Reyes et al., 2017; Manners et al., 2013; Straatsma et al., 2008). We scanned *Populus* patches representing different stages of pioneer woody vegetation growth: seedlings (*sdl*), saplings (*sap*), and young trees (*tr*). From these scans (postprocessed in the same manner described in Bywater-Reyes et al. (2017)), we established an $A_c$ – height relationship (Fig. 2b), from which depth-dependent $A_c$ was extracted for each model run by assigning $A_c$ based on the average bar flow depth from the corresponding no-vegetation scenario.

To test whether overbank (floodplain) vegetation (i.e., beyond the vegetated bar) contributes to flow steering in the main channel and influences the hydraulics of the cutbank–bar region of interest (Fig. 3), we included runs with and without floodplain vegetation for each of the four flows and seven bar vegetation scenarios, resulting in 56 model runs. We represented floodplain vegetation as was observed from airborne LiDAR (see Supplement for more detail). These analyses showed that the hydraulics of the cutbank–bar region of interest (Fig. 3) were insensitive to whether or not floodplain vegetation was present across the range of modelled flow conditions. Therefore the descriptions of hydraulics we present in Results are based only on scenarios varying bar vegetation conditions.

We considered hydraulic ($u$, $v$, $\tau_s$, $\tau_n$) solutions for three cross sections at locations across the bar and cutbank of the channel bend, representing the upstream, midstream, and downstream portion of the bar (Fig. 3). We additionally considered the hydraulics and potential for bed mobility spatially, where the Shields number, $\tau^*$, was used as an indicator of bed mobility:

$$\tau^* = \frac{\tau}{(\rho_s - \rho)gD} \qquad (4)$$

where $\tau$ is boundary shear stress, $\rho_s$ is sediment density, $g$ is acceleration due to gravity and $D$ is grain diameter. We compared the solutions for vegetation runs for each flow to no-vegetation scenarios to evaluate which configurations had the greatest influence on hydraulics.

## 3 Results

The effects of bar vegetation on modelled hydraulics across our study reach are presented here in several ways, including by comparing a no-vegetation case to vegetated cases (density and growth stages), and with respect to variations spatially across the bar at different discharges. For the no-vegetation case, velocity and shear stress were generally highest in the thalweg and lower over the bar (Fig. 4). Downstream velocity ($u$) was generally greater than cross-stream velocity ($v$). The greatest $v$ magnitudes were for the downstream cross section (XS1; Fig. 5c,d). With increasing flow magnitude, both $u$ (Fig. 5b) and $v$ (Fig. 5d) decreased within the thalweg region, but stayed relatively constant over the bar. A similar trend was seen at the mid-



bar cross section (XS2) with $u$ decreasing within the thalweg region as flow magnitude increased, but remaining relatively constant over the bar (Fig. 6). In contrast, $u$ increased within the thalweg region and over the bar with increasing flow (Fig. 7a,b) at the upstream cross section (XS3), whereas $v$ stayed relatively constant (Fig. 7c,d).



5   **Figure 4.** Planview comparison of channel-bend hydraulics (velocity; a–c, and Shields number; c–f) for the $Q_{10}$ no-vegetation (a,d), sparse young trees b,e), and dense seedlings (c,f) runs. Location of cross sections (Fig. 3 shown). Velocity and Shields number are reduced on the bar with increasing size or density of plants, and flow paths within the thalweg and adjacent to the vegetation patch become more concentrated.





**Figure 5. Effect of the vegetated bar (j = 33–75) on the streamwise (*u*; a,b) and stream-normal (*v*; c,d) velocity at the downstream cross section (XS1) for the Q$_2$ (a,c) and Q$_{10}$ (b,d) flows. With increasing plant size (seedling to young trees) and density, *u* is increased and *v* decreased within the thalweg (j = 100). Both *u* and *v* are decreased over the bar, and for the sparse young trees and all dense scenarios increased at the edge of the patch.**





**Figure 6. Effect of the vegetated bar (j = 32–82) on the streamwise ($u$) velocity at the midstream cross section (XS2) for the $Q_2$ (a), $Q_{10}$ (b), $Q_{20}$ (c), and $Q_{100}$ (d) flows. In the thalweg (j = 100), $u$ increases and the maximum shifts toward the left bank. On the bar, velocity is decreased in the patch, and increased at the right edge of the patch.**





**Figure 7. Effect of the vegetated bar (j = 50–65) on the streamwise (*u*; a,b) and stream-normal (*v*; c,d) velocity at the upstream cross**
5 **section (XS1) for the Q₂ (a,c) and Q₁₀ (b,d) flows. In the thalweg (j = 90) and at the head of the bar, *u* is decreased with increasing**
**seedling size and density. For Q ≥ Q₁₀, *v* was decreased (more negative) adjacent to the vegetation patch.**





The manner in which different vegetation densities and growth stages influenced hydraulics varied spatially around the bend. In general, adding vegetation increased velocity within the thalweg and at the edge of the vegetation patch compared to the no-vegetation case, creating concentrated flow paths adjacent to the patch while reducing velocity and shear stress at the head of the bar and within the vegetation patch. The effect of the vegetated bar on channel-bend hydraulics became more pronounced with discharges increasing from the $Q_2$ to $Q_{10}$. Furthermore, sparse vegetation behaved similarly to the no-vegetation scenario for low flows, but had an increasing effect on hydraulics at flows $\geq Q_{10}$.

At the downstream end of the bar (XS1; Fig. 5), vegetation increased the magnitude of downstream ($u$) and cross-stream ($v$ more negative) velocity within the thalweg region, and reduced velocities over the bar. For flows $\geq Q_{10}$, the high-velocity core became more concentrated and shifted away from the bar. This thalweg effect became more pronounced with increasing plant density and plant size, except in the case of dense young trees, which behaved more similarly to the bare bar scenario for the $Q_{10}$ flow. Amplification of thalweg velocities at XS1 was greatest for the dense sapling scenario, with 17 % and 12 % increases in $u$ and $v$, respectively, for the $Q_{10}$. On the vegetated bar, $u$ and $v$ decreased within the vegetated patch, with $u$ values reduced up to 56 % for the sparse young tree scenario, and up to 95 % for the dense scenarios. With increasing plant size and density, the values of $u$ and $v$ at the right edge of the vegetation patch were greater than or nearly equal to that in the thalweg, with a particularly large increase for dense scenarios. Thus, flow velocities were decreased within the patch, increased adjacent to the patch, and were deflected toward the left bank.

At the midstream position (XS2), downstream velocities ($u$) in the thalweg region were greater than at XS1. The impact of the vegetation patch on $u$ for XS2 was pronounced, with $u$ increased up to 30 % within the thalweg and the maximum value of $u$ shifted toward the left bank with increasing plant size, density, and discharge (Fig. 6). Like XS1, the thalweg effect reached a maximum for dense saplings at the $Q_{10}$. As flow increased ($Q_{20}$ and $Q_{100}$), dense trees had the greatest effect on increasing thalweg $u$. On the bar, the effect on $u$ for XS2 was similar to XS. Values of $u$ decreased with increasing size and density of plants, and $u$ increased at the right outer edge of the vegetation patch. Over the bar, $u$ was reduced up to 99 % for the dense scenarios compared to the no-vegetation scenario, and increased at the edge of the patch up to 3300 %. At XS2, $v$ values were small compared to XS1 and XS3 and were relatively insensitive to the presence of the vegetation patch

At the upstream end of the bar (XS3; Fig. 7), an opposite trend in changes in $u$ within the thalweg was observed. With increasing seedling size and density, $u$ was decreased within the thalweg and at the head of the bar, with a maximum reduction in $u$ of 29 % for dense scenarios. For $Q \geq Q_{10}$, $v$ was more positive to the left (70 %) of the vegetation patch and more negative to the right of the vegetation patch (180 % reduced). Within the vegetation patch, $u$ and $v$ were reduced (96 % and 100 %, respectively). Thus, flow was steered away from the vegetation patch.





## 4 Discussion

### 4.1 Impact of vegetation on channel-bend hydraulics

Our results illustrate that vegetation enhances the effects of bars on flow steering, complementing previous work on bend dynamics in the absence of vegetation. Dietrich and Smith (1983) showed that bars steered flow in a manner that forced the

high-velocity core toward the concave bank. They additionally found that flow over the heads of bars resulted in cross-stream components of velocity ($v$) and boundary shear stress ($\tau_n$) directed toward the concave bank. Whiting (1997) hypothesized that convective accelerations arising from flow steering would be most important at low flows, whereas Legleiter et al. (2011) showed that steering from bars continued to be important with increasing discharge. We found that vegetation began to impact channel-bend hydraulics for flows greater than the Q$_2$, when plants began to be inundated. Similar to Abu-Aly et al. (2014),

we found that the influence of vegetation on flow did not decrease with increasing discharge, but stabilized after vegetation-inundation flow depths were achieved. This suggests an initial steep increase in alteration of hydraulics from vegetation from Q$_2$ to Q$_{10}$, with modest changes thereafter.

In general, we found the impact of the vegetated bar on channel-bend hydraulics to increase with both density and size of plants modelled. Dense young trees, however, did not always result in the maximum alteration to channel-bend

hydraulics—particularly for $u$ during the Q$_{10}$ flow. This indicates there may be thresholds whereby increasing density and size of vegetation no longer results in a linear change in hydraulics in some cases.

At the mid- and downstream sections of the channel bend investigated, the presence of dense vegetation increased downstream velocity ($u$) within the thalweg up to 30 % and shifted the high-velocity core toward the cutbank. Vegetation increased the magnitude of cross-stream velocity ($v$) at both the up- and downstream end of the channel bend by increasing

cross-stream flow toward the cutbank at the head of the bar and around the toe of the bar. Positive $v$ values within the thalweg region at the upstream cross section (XS3) indicate, indeed, flow is steered toward the concave bank. By extension, $\tau_n$ is directed toward the concave bank given equation (2). At the head of the bar, flow was additionally slowed within the channel ($u$ decreased), and steered away from the vegetation patch, increasing flow within a side channel adjacent to the bar head and creating concentrated flow paths adjacent to the patch.

Some flume and modelling studies support our results, but others illustrate the variability in hydraulic response to vegetation as a function of channel geometry. For example, Marjoribanks et al. (2017) modelled the effects of vegetation on channel hydraulics for a small (~5 m wide by 16 m long), straight river reach and found downstream ($u$) and cross-stream velocities ($v$) reduced broadly throughout the channel. Here, we found reduction of $u$ and $v$ to vary spatially depending on the location within the channel bend and relative to the vegetation patch. Rominger et al. (2010), working with two reed species

planted on the sandy point bar of a constructed, meandering experimental stream, found that vegetation reduced $u$ values over the vegetated bar, increased them in the thalweg, strengthened secondary circulation, and directed secondary flow toward the outer bank. Another study in the same experimental facility, but using woody seedlings planted on the point bar, also found reduced velocities in the vegetated area of the bar, with the greatest reductions at the upstream end, and strengthening of



secondary circulation, as well as illustrating differences in hydraulic effects as a result of variations in vegetation architecture and density (Lightbody et al., 2012). In another flume study where meandering was simulated in a straight channel by placing dowels representing vegetation patches in alternating locations along the edges of the flume, vegetation reduced velocity within and at the edges of the vegetation patch and increased velocities near the opposite bank, consistent with the results here (Bennett

et al., 2002). However, Termini (2016) considered vegetation's effect on flow in a high-curvature meandering flume and found that vegetation inhibited high shear-stress values from reaching the outer bank, inconsistent with the results found here and in other studies simulating moderate sinuosity channels.

## 4.2 Implications for channel morphology and evolution

The reduction of velocity and shear stress and associated reduction in momentum transfer within the thalweg at the bar head caused by the presence of the vegetated bar would be expected to decrease sediment transport in this region. This may contribute to bar-head maintenance, such that the head of the bar is not eroded. Maintenance of the bar head would be countered by the potential for chute cutoff (van Dijk et al., 2014) or channel switching that may result because of concentrated flow paths.

15         The production of a low-velocity region over the vegetated bar would be expected to increase sediment deposition, especially of fine sediment, on the bar, consistent with flume and field observations. Elevated sediment deposition within patches of woody seedlings, with variations depending on plant characteristics, has been documented in meandering (Kui et al., 2014) and straight (Diehl et al., 2017b) flumes. Gorrick and Rodríguez (2012), working in a flume in which vegetation patches were simulated with dowels, documented elevated fine-sediment deposition within the patches (Gorrick and

Rodríguez, 2012). Zones of fine sediment deposition on bars associated with roughness from vegetation or instream wood can in turn create sites for plant germination and seedling growth (e.g., Gurnell and Petts, 2006). At our field site, we developed a grain-size patch map (Buffington and Montgomery, 1999) on the vegetated bar, which illustrates an obvious correlation between sandy patches and the location of trees that have experienced several floods (Fig. 8). Increased deposition of sediment on the bar would also contribute to bar building, imposing an additional feedback as topographic steering from the bar is

enhanced.



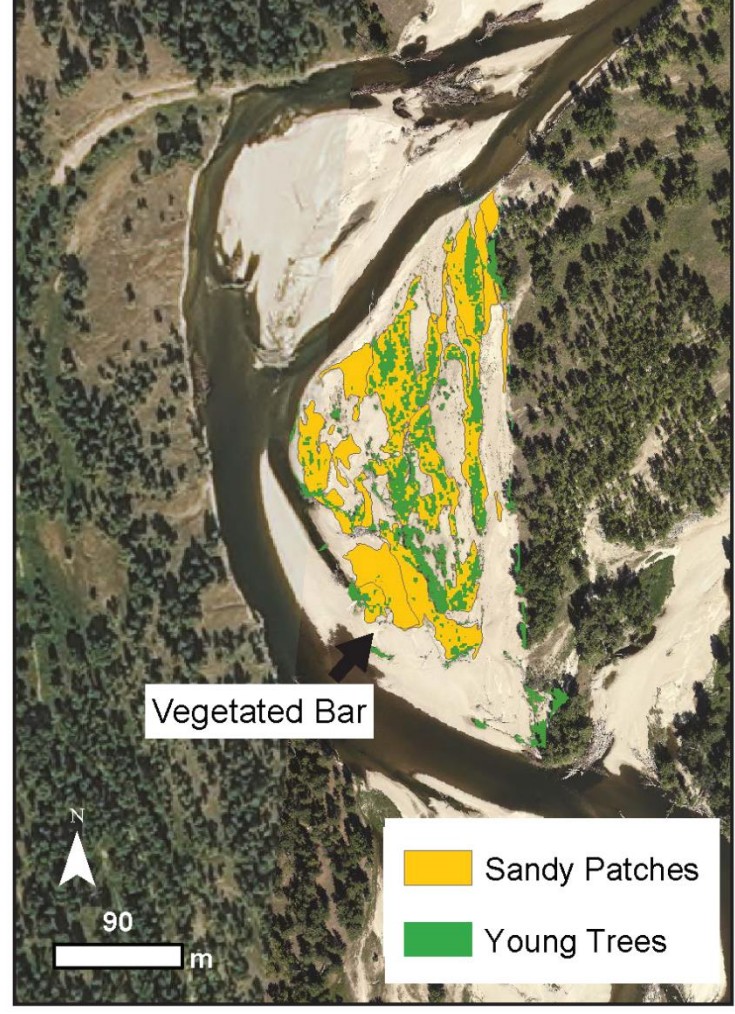

**Figure 8. The vegetated bar, showing the spatial co-occurrence of sandy grain size patches and young trees (extracted from aerial imagery).**

The increase in velocity and shift of the high velocity core toward the cutbank combined with low velocities within the vegetation patch would create a large velocity gradient across the channel. A larger velocity gradient within the thalweg compared to over the bar would be expected to alter the dynamics of bank erosion. As a simple rule, bank erosion rate, $\dot{n}$, according to Parker et al.'s (2011) HIPS model is proportional to an erosion coefficient, $k$, and half the streamwise velocity difference between the two banks, $\Delta u$:

10                              $\dot{n} = k\Delta u$        (5)



The parameter, *k*, represents the material cohesion and vegetation root properties that control bank erosion and varies between $10^{-8}$ and $10^{-7}$ (dimensionless). Thus, for an assumed *k*, vegetation-induced velocity gradients across the channel are expected to alter bank erosion rates.

Vegetation "pushing" flow toward the outer bank is analogous to "bar push" (Allmendinger et al., 2005; Parker et al., 2011), whereby a rapidly accreting point bar may cause erosion at the outer bank (Eke et al., 2014; van de Lageweg et al., 2014). This increase in bank erosion would be countered by deposition of fine sediment on the bar resulting from the vegetation-induced reduction in velocity in this region, that may in turn induce addition "push" through bar building (e.g., Eke et al., 2014). Coarse bank roughness counters this effect, pushing the high velocity core back toward the center of the channel (Gorrick and Rodríguez, 2012; Thorne and Furbish, 1995). The balance between erosion of the bank and deposition on the bar would thus dictate whether net erosion or net deposition within the active channel occurs, inducing changes in channel width (Eke et al., 2014), and altering channel morphology.

Width-to-depth ratios higher and lower than expected based on at-a-station hydraulic geometry have been reported for vegetated channels (Corenblit et al., 2007). Initial riparian forest development may result in a decrease in width-to-depth ratio as formerly bare banks are vegetated and increase bank cohesion, preventing bank erosion from widening channels (Métivier and Barrier, 2012) such that meanders (Eaton and Giles, 2009) and alternate bars emerge (Kleinhans, 2010). For channels characterized by vegetated banks and meandering planforms, differences in width have been observed based on floodplain and bank vegetation type, with floodplains composed of herbaceous vegetation associated with narrower channels compared to those composed of woody vegetation (Allmendinger et al., 2005; Hession et al., 2003; Jackson et al., 2015). It is unclear what would cause this relationship, since bank strength increases with rooting depth (Eaton and Giles, 2009), which is greater for woody vegetation compared to herbaceous vegetation (Canadell et al., 1996). Our site has woody vegetation on banks and floodplains, and has both bars with abundant vegetation (Bar 1) and those relatively free of vegetation (Bar 2, Bar 3; Fig. 1). Comparison of the morphology of the vegetated bar at the site to two others with very little vegetation (2012 topography; Fig. 9) shows that the bars had similar widths, but the vegetated bar (Bar 1) had a deeper thalweg. This may be a manifestation of increased, concentrated velocity and shear stress in this region. This suggests the vegetated bar had a smaller width-to-depth ratio compared to the others, inconsistent with the notion that floodplains composed of herbaceous vegetation are associated with narrower channels compared to those composed of woody vegetation (Allmendinger et al., 2005; Hession et al., 2003; Jackson et al., 2015).



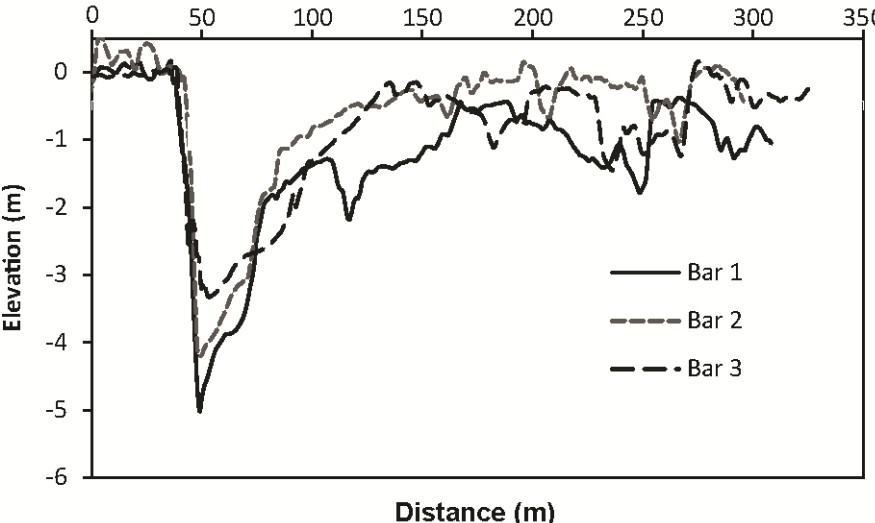

**Figure 9. Cross sections for the vegetated bar (Bar 1), and two others (locations shown in Fig. 1). The vegetated bar (Bar 1) has a deeper thalweg compared to the other two bars, but similar widths. Note the horizontal axis for Bar 3 has been reversed for comparison of geometry.**

However, the width-to-depth ratio of a channel should adjust depending on the outcome of bars and vegetation "pushing" banks, versus bar accretion. On the one hand, vegetation may decrease width-to-depth ratios from a combination of increased bank strength and scouring deeper thalwegs because of concentrated flow around the bend. On the other, concentrated streamwise flow paths at the head of the bar combined with a shift in the high-velocity core toward the cutbank, large differential in cross-stream velocities, and cross-stream accelerations would tend to increase bank erosion at the mid- and downstream end that may be accompanied by bank-pull bar building.

**5 Conclusion**

The presence of a vegetated bar in a gravel-bed river altered both streamwise and cross-stream components of velocity vectors for overbank flows, with an increasing effect with discharge and both plant density and size. Vegetation steered flow away from the vegetated bar, creating concentrated flow paths in surrounding low-elevation side channels and a low-velocity region over the vegetated patch. Flow was slowed at the apex of the bar, and increased within the thalweg around the bend. These changes in hydraulics are expected to increase fine sediment deposition on the bar, potentially creating hospitable sites for vegetation recruitment, and increasing bank erosion that is dependent on cross-stream velocity gradients. This pattern would tend to reduce cross-stream sediment transport at the bar head, but increase it around the remainder of the bend.



Our analysis simplified vegetation drag by assuming rigid cylinders, and recent research has suggested this is most accurate for dense vegetation (Vargas-Luna et al., 2016). Future research directions include refining how vegetation drag is represented for sparse vegetation, as well as quantifying changes in drag that result from streamlining and reconfiguration during inundation (Aberle and Järvelä, 2013; Boothroyd et al., 2017; Nepf, 2012; Västilä et al., 2013; Västilä and Järvelä, 2014; Whittaker et al., 2013).

Following the patterns of hydraulics and forces, we would expect vegetation to change the morphodynamic evolution of channels with vegetation pushing flow in a manner previously only attributed to bars, and may explain the enigmatic observation that reaches characterized by woody vegetation are wider than those with herbaceous vegetation. Subsequent bank retreat may induce bar building, which would be accelerated by fine-sediment deposition within the vegetation patch. This feedback would induce additional topographic steering from the presence of the bar. We have characterized a mechanisms by which channels with vegetated bars may evolve different morphologies and rates compared to those without, thereby contributing to understanding of ecogeomorphic feedbacks in river-floodplain systems (Gurnell, 2014) and of how life influences landscapes (Dietrich and Perron, 2006).

**List of terms**

$A_c$ = vegetation frontal area (m$^2$)

$A_S$ = frontal area of stems (m$^2$)

$C_d$ = channel drag coefficient

$C_{d,v}$ = vegetation drag coefficient

$D$ = median grain size (m)

$F_D$ = vegetation drag (N m$^{-2}$)

$g$ = acceleration due to gravity (m s$^{-2}$)

$k$ = bank erosion coefficient

$u$ = streamwise component of velocity (m s$^{-1}$)

$\bar{U}$ = depth-averaged velocity (m s$^{-1}$)

$U_x$ = x component of velocity in Cartesian coordinate system (m s$^{-1}$)

$U_y$ = y component of boundary velocity in Cartesian coordinate system (m s$^{-1}$)

$U_c$ = approach velocity (m s$^{-1}$)

$U_m$ = cross-section mean velocity (m s$^{-1}$)

$v$ = stream-normal component of velocity (m s$^{-1}$)

$\rho$ = density of water (kg m$^{-3}$)

$\rho_s$ = density of sediment (kg m$^{-3}$)

$\tau$ = boundary shear stress (N m$^{-2}$)

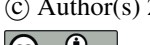

$\tau^*$ = Shields number

$\tau_s$ = stream wise component of boundary shear stress (N m$^{-2}$)

$\tau_n$ = stream-normal component of boundary shear stress (N m$^{-2}$)

$\dot{n}$ = bank erosion rate

**Data availability**

Please inquire with Missoula County for aerial LiDAR. Ground-based LiDAR is available at https://tls.unavco.org/projects/U-026/; Bitterroot Site 1 DOI: 10.7283/R34M07; Bitterroot Site 2 DOI: 10.7283/R30W61, Bitterroot Site 3 DOI: 10.7283/R3W62P. FaSTMECH solver files are available upon request.

**Author contribution**

S. Bywater-Reyes and A.C. Wilcox designed the modelling experiment. R.M. Diehl contributed to updating FaSTMECH code to account for vegetation drag. S. Bywater-Reyes carried out field work and model construction, calibration, and implementation. S. Bywater-Reyes wrote the manuscript with contributions for all co-authors.

**Acknowledgements**

This research was funded by the National Science Foundation (EAR-1024652, EPS-1101342) and EPA STAR Graduate Fellowship. We thank Mark Reiling, Philip Ramsey and MPG Ranch for access to the Bitterroot site. We thank Missoula County for providing LiDAR. We thank Sarah Doelger and UNAVCO, Austin Maphis, Katie Monaco, April Sawyer, and John

Bowes for assistance in the field. A special thanks to Carl Legleiter for sharing his scripts and Richard McDonald, Gregory Pasternack, Daniele Tonina, David Machač, Nicholas Silverman, and Doug Brugger for modelling and scripting tips. The Mexcgns-Matlab scripts used in this study are available upon request of the first author.

**Supplement**

Supporting experimental procedures can be found in the supplement

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
