# Peer review of "The influence of a vegetated bar on channel-bend flow dynamics"

_Earth Surface Dynamics, 2017_

## Referee Comment (RC1) · Anonymous Referee #1 · 27 Oct 2017

General comments: I think this is an interesting study. My main concerns are that the introduction needs to include more of a literature review on what is already known about vegetation effects on flow within meander bends because many of the results presented (at least in terms of overall vegetation effects, perhaps not effects of density/vegetation stage) here are similar to previous laboratory studies. I also think that much of the discussion is highly speculative, which can be fine, but often the speculation exceeds the amount of data needed to be presented to support the suggested hypotheses.

Specific comments: Page 2, line 2: I would argue that vegetation impacts on altering the flow velocity itself (e.g. mean flow velocities, velocity profiles) as stated here have been very well studied. Flow steering, in parentheses, by vegetation has also received

attention but none of the studies that have investigated this are cited here. For example, in the discussion you review many of the laboratory studies that have investigated flow in meander bends with and without vegetation. These studies already demonstrate that vegetation can steer flow toward the outer bank, which is one of the main points of this paper. It seems like these studies should be reviewed here to highlight what is already known, and what is not known that your study is trying to address. What is this study addressing that has not been previously answered? Right now the motivation for why this work is needed is not coming through in the literature review.

Page 3, line 18: A bankfull Shields number for a gravel bed river of 0.01 would imply there is no sediment transport at bankfull flow given that the critical Shields stress is typically greater than 0.03 (Buffington and Montgomery, 1997) for these rivers. It seems somewhat unlikely that there is no transport at bankfull? In addition, cross stream and downstream shear stresses, as well as Shields stresses, are mentioned in the methods but I don't ever recall them being quantified in the results or discussion (except a map of Shields stresses in Figure 4). Why are they brought up in the methods? How did you distribute the vegetation on the bar? Did it cover the entire bar? Was it only in a certain zone where you expect vegetation to establish? The results that you obtain seem like they will be highly dependent on this chosen location and extent of the vegetation patch. For example, on Page 14, line 15: It is stated that the u and v velocities on the right side of the downstream of the vegetated bar (Figure 5) approach or equal those in the thalweg and that this is more pronounced with vegetation density. This is where the effect of vegetation patch distribution comes into play, if the vegetation patch did not extend to the channel bank then this is what one might expect. How much of this result is driven just by the lack of vegetation between the bar and the channel wall (I am assuming this is what you modeled)? Is such a complete break in vegetation likely to occur in nature?

Page 14, line 25: v values are not shown for XS2, which is near the bend apex and it is stated that the presence of vegetation did not really affect the v velocities. If the case
is being made in the discussion that vegetation will change bank scour and meander migration, doesn't this result imply that at the bend apex, although the high downstream velocity core shifts toward the left bank, the actual direction of the flow is not deflected more toward this bank with the presence of vegetation? What does this mean for bank scour at the bend apex?

Page 16, line 15-16: A low velocity region on the bar would imply lower sediment fluxes, but would not necessarily imply sediment deposition, which is the divergence of the sediment flux. Sediment deposition would only occur if the vegetation did not reduce the steering of sediment (sediment supply) into the patch itself. Given that you show that sometimes flow is steered away from the bar on the bar sides, it seems likely that the vegetation will also impact how much sediment enters the bar, and therefore whether deposition occurs.

Page 18, lines 12-27. Much of this discussion does not seem directly related to any of the results presented above, and in particular the comparisons of three bars with/without vegetation to state that there is a difference in w/d and channel narrowness is highly speculative. No w/d ratios are provided for the bars to demonstrate this. I am not clear how only three cross-sections at one study site with no variation in vegetation type (just vegetated vs. not vegetated) can be used to infer that floodplains with herbaceous vegetation may not have narrower channels than those with woody vegetation. Further, although the vegetated bar does have a deeper thalweg, it seems to often have lower elevations on the bar, which is contrary to the earlier discussion that vegetation would cause higher amounts of sediment deposition on bars.

Figure 8 and associated text: Although there are definitely locations where sand is collocated with vegetation, there are also locations where sand deposits are not located around vegetation, or that vegetation patches lack sand deposits. Can you provide more quantitative data to show that sand and vegetation are correlated such as % of sand patches within a certain distance of vegetation or something similar?

Technical questions: Page 1, Lines8-9: You mention alternating bars and vegetation but then discuss bend hydraulics and forces. What kind of forces are you discussing here? Alternating bars do not have to be associated with bends and it is not clear how the second half of the sentence is related to the first. The rest of the abstract seems to be geared toward a bar in a bend, which would normally be called a point bar? This comment is relevant throughout the paper where bar is used. It might be better to be more specific here about what kind of bar you mean.

Page 1, Line 11: "with and without varied vegetation parameters" is not clear here. Are you eliminating the parameters or the vegetation itself? What kind of parameters?

Page 3, line 17: I don't know if the condition of "few upstream dams" implies that flow and sediment supply are relatively unregulated. You can have just one dam upstream that can completely alter the hydrology and sediment supply downstream; it is just not the number of dams that control these parameters but how the dams are operated. Do the dams not alter the flow? Does sediment bypass the dams?

Line 9, page 8: How was $U_m$ determined? At a cross-section upstream of the vegetation that is free from the vegetation influence?

Lines 4-7, page 9: The dense vegetation case is two orders of magnitude higher than the sparse case but both are averages on the same bar. It seems like these two averages should be the same if the average of local densities is representative of what would occur at the scale of the entire bar. Is this partly driven by the scale over which the measurements were taken, in that the 20 stems/m2 value is a local measurement and therefore likely to be higher? Is 20 stems/m2 a realistic value of stem density for an entire bar; is such an average density found in real rivers over the spatial scale of a bar?

Line 11, page 9: If you are using the flow depth based on the model run without vegetation to assign Ac, won't this skew your Ac values because the actual flow depths will likely be higher in the presence of vegetation? Also in Figure 2c, there are many

**ESurfD**
lines but only three stages of vegetation growth, and it is not possible to tell which relations were actually used in the model.

Equation (4): What grain size is used and did the grain size spatially vary in the stream, and in this calculation?

Other methods: How were the stage and nearby discharge used to calculate Q? Why is stage needed and not just a drainage area correction? How many topographic cross sections were measured in the channel, what was the spacing of the cross-sections and what was the actual point density of the DEM in the channel? No information is provided as to how water surface was measured, where it was measured and how many data points were measured for a given flow? A 18 cm RMSE for flow depth could be pretty large, depending on the flow depth magnitude. How large were water surface elevation and velocity RMSE relative to the flow depths and velocities measured in the channel? How many measured/log profile velocities were compared to the modeled velocities to obtain the RMSE? How good were the log profile fits to the measured velocities; are there large errors in what you are assuming to be measured depth-averaged velocities?

Figure 5 It would help to have the direction of the v velocity (which way is negative) noted on the figure or in the caption. There really does not seem to be any change in the v velocity in the thalweg for the Q2 flow, contrary to what is stated in the figure caption.

Page 15, line 15: Can you give an example of where dense trees do not have the maximum impact on the flow velocity as stated here? I don't remember this being discussed in the results. Also, you have modeled the drag coefficient for vegetation as being a constant with vegetation density or plant size, but studies on vegetation have shown that this coefficient can change with vegetation spacing. How might this impact your results?

Page 15, line 20: It is stated that vegetation increased the magnitude of v at the downstream end of the channel bend in the thalweg. In the associated figure, v either did not really change with vegetation or decreased with vegetation, implying instead that cross stream flow was not necessarily directed more toward the cutbank in this cross section. Secondary circulation should be present in all of these cross-sections and therefore, the direction of the v component of velocity will likely depend on the vertical position in the flow column. So I am not sure how much information the depth-averaged v provides in terms of the process of bank erosion? Perhaps you can comment on this.

Page 15-16, lines 30-2: What is similar or different in these studies in the outdoor lab from your study and why are there differences in the studies? The discussion on what is similar or different is somewhat vague and do not really include hypothesizes why you might see different results in your model.

Page 16, lines 10-11: It is stated that the flow velocities and shear stresses in the thalweg in the upstream cross-section are reduced with vegetation but in Figure 7, u is reduced but v is increased with vegetation and it is therefore not clear what will happen to shear stress (and sediment transport and erosion), which is not shown.

Conclusion: Please see my earlier comments above about whether vegetation will cause fine sediment deposition. Certainty this is what others have found, but I am not sure that the data you present allow you to say that deposition will occur unless you assume that the sediment input to the bar is not changed. It is not clear why cross-stream sediment transport would be reduced by the changes in flow that are mentioned, can you provide more information on this? I think the statement that "previously only attributed to bars" is not entirely true given that previous studies in meander bends have shown that vegetation can direct the flow toward the opposite bank.

Figure 7: In the caption it is stated that v decreased by becoming more negative but changing from a low negative value to a higher negative value means that the velocity actually increased because the negative sign only denotes direction. I think that you mean less negative or at least that is what the figure appears to show to me but I can't

really tell what part of "adjacent to the patch" you are referencing here—on the left or the right side?

Comments on supplemental information:

You alternate between u and U being velocity at a given elevation above the bed. I think you should pick one.

It is not clear to me why you used the log profile fits instead of just using the measured velocity at 0.37h. Did you not always have this measured data point because of data exclusion near the water surface? It seems like using the measured values, if possible, would lead to less uncertainties than fitting a profile and then calculating a mean value from those fits. Or do you think there are large uncertainties in a given data point, making the profile fit more reliable? How many data points were used in the velocity profile fits?

Lines 44-46: I find it slightly confusing what is being compared in this sentence. RMSE of the modeled values calculated either using the log profile or the extrapolated velocity values? Does FASTMECH assume a log profile in its calculations of mean velocity? If so could this partly explain why you obtained lower RMSE when using the log profile instead of using the extrapolated values to the water surface?

Line 64 Is this a standard error or deviation?

---

## Referee Comment (RC2) · Anonymous Referee #2 · 6 Nov 2017

This is an interesting study, which examines the impact of different vegetation types and densities on flow through a channel with a vegetated bar. The topic is relevant and the work builds on a significant literature in this area. While the work seems rigorous and of good quality, there are some details of the methodology that would benefit from clarification. Furthermore, the data could be better presented to improve clarity.

Major Comments:

Representation of vegetation: The authors raise the issues regarding the use of roughness coefficients for representing vegetation. Accordingly, they adopt a much more suitable drag-based approach. However, there are still potential limitations with this approach. In particular: the parameterisation of drag coefficient, the distribution of drag elements in space and the assumption of a logarithmic profile may represent sig-

nificant limitations of the study and could receive more attention in the text (see specific comments below)

Methods: There are a number of details regarding the numerical methodology which are currently not presented, but which may have a significant impact on the results (e.g. average drag force equation, grid size & type, relative errors, approximate depths, delineation of bar).

Figures & Data: Figure 2b could be presented more clearly. Figures 5-7 could be made clearer, but also some data is referred to which is not present in these figures (higher Q values for XS1 &3).

Specific comments: Pg 5 Ln 5: Is A_S defined? Appears in supplementary data, but I'm not sure it is defined in the main text?

Pg 6 Ln 12: What was the grid size used in the simulation? Was it constant for the whole domain? Was bank (wall) shear stress included too? (i.e. cells with wall boundaries too).

Pg 6 Ln 20: In Table 1 it would be helpful see the relative magnitude of errors. Errors of 0.18m in WSE and 0.36m/s in velocity seem large, but may not be relative to the mean values? Table 1 does also not provide a comprehensive overview of the calibration. E.g. which different values were used for C_d? What was the sensitivity to this value? The two LEV values are an order of magnitude apart, were any other values in between tested? What was the rationale for picking these values? Also, the table seems to suggest that a model without any vegetation performed better than the model with vegetation?

Pg 7 Ln 11: These relaxation figures mean very little out of context. Please provide brief explanation of which variables they correspond to.

Pg 7 Ln 15: Why were you unable to maintain a curvilinear grid? This is unclear. Which nodes overlapped and why? Was the model run in Cartesian grid? Section 2.2 seems

to suggest it was curvilinear (Pg 6, Ln 5). If values were converted between grids, how was this done, i.e. interpolation methods, grid sizes etc.

Pg 8 Ln 7: Presumably the model uses an equation in terms of drag force per unit volume? It would be useful to include the exact form here.

Pg 8 Ln 10: I agree with the authors that $C_D=1$ is a common first-order approximation, and probably does an ok job for the lower section of the plants where objects are likely to be cylindrical. However, for trees, with complex foliage I would expect this assumption to be less accurate. Therefore, it might be worth reflecting on the accuracy of the model at different discharges

Figure 3: How was the vegetated bar delineated? Current vegetated extent?

Pg 8 Ln 13: If I am correct, a height-dependent value of A is used (from Figure 2). However, regardless of depth, the near-bed vegetation geometry will not change. Therefore, in terms of defining near-bed processes linked to sediment transport, I wonder what the impact is of changing $A_c$ as depth increases, given that this impact may only be significant towards the top of the flow? Above a certain height, does the effect of area on bed-processes diminish?

Pg 8 Ln 12: How does the grid resolution compare with the stem density? Are the effects of a single stem artificially 'smeared' over many stems? If so, particularly for low vegetation densities, the flow patterns may not correspond well with single, isolated large area blockages, which will have a very different impact to wide-spread small blockages.

Pg 8 Ln 13-14: The flow will typically not be logarithmic where there is vegetation present. Therefore, what errors does this assumption introduce? Are the results valid?

Pg 9 Ln 5: 20 stems per square metre seems very dense for saplings and trees? Also, for such densities, is it still valid not to consider the mass blockage effect of the vegetation?

Pg 9 Ln 32: Decreasing velocities in the thalweg is surprising –but seems to correspond to additional flow along a separate channel to the right of the vegetated bar? It seems this is quite an important aspect which affects other results too (e.g. flow deflection into this channel for certain vegetation conditions). This could be made clearer within the discussion which frames the problem as a simple channel bend with vegetated bar.

Pg 9 Ln 32: Are the observed decreases/increases in velocity significant with respect to uncertainty/error?

Figures 5-7: These graphs are not easy to read. I wonder if colour could be used in addition to line style, or results separated for density & type? Furthermore, it is unclear why lateral velocities are not reported for XS2?

Pg 14 Ln 9: Would be helpful to show the data for each XS for Q>10, not just XS2.

Pg 14 Ln 16-17: As mentioned above, it seems the side channel to the right of the patch plays an important role in conveying discharge, particularly for higher Q values. Is this process more important than channel bend processes?

Pg 15 Ln 8-10: I agree that results show that the impact of vegetation increases with Q, but I do not think results show that the vegetation begins to impact on channel-bend hydraulics for Q>Q2. It seems to me that even at Q=Q2 there are significant differences in velocity distributions that may, over a long period cause significantly different channel morphology?

Pg 15 Ln 16: I do not think the results show any evidence of 'linear' trends?

Pg 16 Section 4.2: It would be good to quantify the correlation between sediment and vegetation, beyond the visual observation in Figure 8. Also, these patterns demonstrate the limitation of assuming constant vegetation density across the bar as mentioned earlier.

Pg 18 Ln 21: The authors mention the presence of bars with vegetation/no vegetation. This study investigates the difference of plant type (age) but this in itself is related

**ESurfD**
[Figure]

to channel morphology (e.g. plant succession over time) and flood discharges (e.g. destroying plants or creating new bars). It would be interesting to think about how the model could be developed to introduce different vegetation types, depending upon bar age, etc.

---

## Editor Comment (EC1) · JM Turowski (Editor) · 14 Nov 2017

Dear authors,

we have received two reviews for your paper, both of which are detailed and comprehensive. From these reviews and my own reading of the manuscript, I think that currently the major short-coming lies in the statement of the research gap. Although you have given the objectives, it is currently unclear which open research question you trying to answer and how this fits into the existing literature. Both reviewers mention that you have overlooked relevant published papers. I suggest that you identify a research gap through a detailed literature review in the introduction and state it clearly together with the research question in the final paragraph of the introduction. The

specific objectives should follow out of this research question. In the discussion and conclusion, you can pick up the question and objectives and place your new insights into the body of the existing literature.

The reviewers make a large number of other points, and there are open questions on methods, results and discussion. Please take all of these points seriously when revising the paper.

All the best and looking forward to your revised manuscript, Jens Turowski

---

## Author Comment (AC2) · 9 Dec 2017

AC: We thank the reviewers for their insightful comments, to which we have responded in detail below. Major revisions to the paper include A) reframing the introduction and motivation of the research by synthesizing what we know about vegetation and channel bends from the literature; B) clarifying details concerning methodology by adding this information to the main text or referring to the Supplement, where much of the details were already housed; C) more explicitly stating assumptions of modeling approach; and D) revising the discussion by deleting portions that bordered speculative

(fine-sediment deposition and channel geometry in vegetated channels) and adding in additional insights related to ecogeomorphic feedbacks and chute channels on vegetated point bars. We believe the manuscript is clearer and more focused. Thanks for your consideration.

Anonymous Referee #2

R2C1: This is an interesting study, which examines the impact of different vegetation types and densities on flow through a channel with a vegetated bar. The topic is relevant and the work builds on a significant literature in this area. While the work seems rigorous and of good quality, there are some details of the methodology that would benefit from clarification. Furthermore, the data could be better presented to improve clarity.

AC1: We have clarified methodology questions and will improve figures, in response to specific reviewer suggestions, before final resubmission.

R2C2: Major Comments: Representation of vegetation: The authors raise the issues regarding the use of roughness coefficients for representing vegetation. Accordingly, they adopt a much more suitable drag-based approach. However, there are still potential limitations with this approach. In particular: the parameterisation of drag coefficient, the distribution of drag elements in space and the assumption of a logarithmic profile may represent significant limitations of the study and could receive more attention in the text (see specific comments below)

AC2: We recognize that our modeling does not fully represent the complexities of field-based vegetation and flow conditions; we have added or revised text in several locations to highlight our assumptions and/or limitations, including a paragraph in the Discussion (end of 4.1) explicitly discussing these issues.

R2C3: Methods: There are a number of details regarding the numerical methodology

which are currently not presented, but which may have a significant impact on the results (e.g. average drag force equation, grid size & type, relative errors, approximate depths, delineation of bar).

AC3: Both reviewers requested clarifications on modeling details. We have added text to address these details in both the main text and the Supplement (in some cases, the information requested by reviewers was in the Supplement in the original version). We have added a sentence pointing readers to the Supplement early in Methods. In some cases, we have moved details that were previously in the Supplement to the main text, in response to review comments, but some details we consider more suitable for the Supplement.

R2C4: Figures & Data: Figure 2b could be presented more clearly. Figures 5-7 could be made clearer, but also some data is referred to which is not present in these figures (higher Q values for XS1 &3).

AC4: The revised manuscript will include revisions to increase the clarity of several figures. Specifically, we will add the average curve for each seedling size to Figure 2. For Figures 5-7 we will add colors and include all figure combinations; important examples will be in the main text and additional combinations, of which there are many, will be presented in the Supplement.

R2C5: Specific comments: Pg 5 Ln 5: Is A_S defined? Appears in supplementary data, but I'm not sure it is defined in the main text?

AC5: Changed to Ac to reduce confusion/simplify

R2C6: Pg 6 Ln 12: What was the grid size used in the simulation? Was it constant for the whole domain? Was bank (wall) shear stress included too? (i.e. cells with wall boundaries too).

AC6: We have added details on grid resolution to the main text (2.5 x 2.5 m cells for calibration runs, 5 x 5 m cells for remaining runs); they are also in the Supplement.

The grid size was constant for the whole domain (although as noted, differed between calibration and other runs). We added a reminder in the main text to point readers to the Supplement. Wall stresses were not calculated. These should be negligible in the channel modeled here, where width » depth.

R2C7: Pg 6 Ln 20: In Table 1 it would be helpful see the relative magnitude of errors. Errors of 0.18m in WSE and 0.36m/s in velocity seem large, but may not be relative to the mean values? Table 1 does also not provide a comprehensive overview of the calibration. E.g. which different values were used for C_d? What was the sensitivity to this value? The two LEV values are an order of magnitude apart, were any other values in between tested? What was the rationale for picking these values? Also, the table seems to suggest that a model without any vegetation performed better than the model with vegetation?

AC7: We added more detail to Table 1 and the Supplement concerning WSE and Åł calibration. We added details to the text concerning the range of LEV and Cd values tested. The model with vegetation for Q2 (453 m3/s) did perform slightly better in terms of WSE, but by a minimal amount. We do not have Åł measurements for this flow.

R2C8: Pg 7 Ln 11: These relaxation figures mean very little out of context. Please provide brief explanation of which variables they correspond to.

AC8: Added clarifying text ("FaSTMECH uses relaxation coefficients to control changes in a parameter between iterations (Nelson, 2013). Relaxation coefficients were set to 0.5, 0.3, and 0.1 for ERelax, URelax, and ARelax, respectively, through trial and error.")

R2C9: Pg 7 Ln 15: Why were you unable to maintain a curvilinear grid? This is unclear. Which nodes overlapped and why? Was the model run in Cartesian grid? Section 2.2 seems to suggest it was curvilinear (Pg 6, Ln 5). If values were converted between grids, how was this done, i.e. interpolation methods, grid sizes etc.

AC9: We added the text "We were unable to maintain a curvilinear, channel-fitted grid

(nodes overlapped) so we projected our Cartesian coordinate flow solution output to the nearest grid cell of a curvilinear grid (2 by 2 average grid resolution) covering the main channel, and converted the associated output to streamwise and stream-normal values with a rotation matrix. A piecewise Cubic Hermite Interpolating Polynomial algorithm was applied to reduce artifacts from the transformation" to the Supplement.

R2C10: Pg 8 Ln 7: Presumably the model uses an equation in terms of drag force per unit volume? It would be useful to include the exact form here.

AC10: Added detail that drag is averaged over vegetation polygons. Because the model is 2D, drag force is per bed area, not volume.

R2C11: Pg 8 Ln 10: I agree with the authors that C_D=1 is a common first-order approximation, and probably does an ok job for the lower section of the plants where objects are likely to be cylindrical. However, for trees, with complex foliage I would expect this assumption to be less accurate. Therefore, it might be worth reflecting on the accuracy of the model at different discharges

AC11: We added a paragraph in the Discussion (end of 4.1) discussing these issues.

R2C12: Figure 3: How was the vegetated bar delineated? Current vegetated extent?

AC12: Vegetated bar was delineated based on current mapped vegetation extent (Fig. 1), as indicated in the main text and the Supplement. (one of our responses to Referee 1 also addresses delineation of the vegetated bar)

R2C13: Pg 8 Ln 13: If I am correct, a height-dependent value of A is used (from Figure 2). However, regardless of depth, the near-bed vegetation geometry will not change. Therefore, in terms of defining near-bed processes linked to sediment transport, I wonder what the impact is of changing A_c as depth increases, given that this impact may only be significant towards the top of the flow? Above a certain height, does the effect of area on bed-processes diminish?

AC13: This is correct, a height-dependent value of frontal area is used, from Fig. 2

(which we will revise for clarity; we have also made minor revisions to the caption for clarity). As shown in Fig. 2, the height dependence of Ac is most important for depths between ∼0.2 and 1 m, with variations among growth stages, and diminishing effects at greater heights (Fig. 2). We agree that near-bed processes most linked to sediment transport are not fully captured by this approach. Given our focus on hydraulics, rather than near-bed sediment transport processes, we consider our approach to be adequate. Indeed, we consider using field measurements of vegetation structure with ground-based LiDAR to determine frontal area and variations with height, for different growth stages, and incorporation of height / depth-dependence of frontal area into modeling, to be an advance over standard modeling practices and a strength of our study.

R2C14: Pg 8 Ln 12: How does the grid resolution compare with the stem density? Are the effects of a single stem artificially 'smeared' over many stems? If so, particularly for low vegetation densities, the flow patterns may not correspond well with single, isolated large area blockages, which will have a very different impact to wide-spread small blockages.

AC14: We have added details on grid resolution to the main text; they are also in the Supplement. Stem density is used to calculate projected vertical frontal area of vegetation and vegetation form drag (eq. 1). Our intent here is not to represent the effects of vegetation at all scales, but rather to assess two end-member density and vegetation drag scenarios. We recognize the complexity of vegetation affecting hydraulics at multiple scales as a function of patch configuration. We have treated these topics in other papers (see response to comment below). We reference Vargas-Luna et al. (2015a) in that representing vegetation as cylinders averaged over an area works best for dense vegetation.

R2C15: Pg 8 Ln 13-14: The flow will typically not be logarithmic where there is vegetation present. Therefore, what errors does this assumption introduce? Are the results valid?

AC15: We recognize that vegetation will disrupt logarithmic velocity profiles, and we agree that a complete representation of vegetation effects on the velocity profile is a worthy goal, albeit one that we consider beyond our scope. We added text more explicitly recognizing the limitation of assuming a log velocity profile ("The model assumes a logarithmic velocity profile, although we recognize this is an over-simplification of how factors such as vegetation submergence alter velocity profiles (e.g., Manners et al., 2015)." In general (including via revisions in response to comments here) we have sought to be transparent about the limitations of our modeling approach, and to emphasize results and insights that we consider valid even in light of those limitations.

R2C16: Pg 9 Ln 5: 20 stems per square metre seems very dense for saplings and trees? Also, for such densities, is it still valid not to consider the mass blockage effect of the vegetation?

AC16: Densities of 20 stems / m2 are indeed dense, but are consistent with literature values; we have added references. Furthermore, our objective is to investigate end-member cases. With respect to the second part of the comment, regarding mass blockage effect, we agree that this could be an important effect for larger-diameter plants. For the size (diameter) of plants in our field site, even at the high densities considered here, we do not expect plants to act as collective bodies with mass blockage effects. We have thought extensively about the relationship between vegetation morphology and organization on hydraulics. In Bywater-Reyes et al. (2017, JGR-ES), we use terrestrial laser scans of woody seedlings to measure roughness density, blockage effects, and implications for hydraulic structures. In Manners et al. (2015, JGR-ES) and Diehl et al. (2017, ESPL), we measure (in a flume) how woody seedlings differentially affect hydraulics and topography depending on whether they are organized individually or in patches.

R2C17: Pg 9 Ln 32: Decreasing velocities in the thalweg is surprising –but seems to correspond to additional flow along a separate channel to the right of the vegetated bar? It seems this is quite an important aspect which affects other results too (e.g.

flow deflection into this channel for certain vegetation conditions). This could be made clearer within the discussion which frames the problem as a simple channel bend with vegetated bar.

AC17: We have added text to the Discussion (section 4.2) about the low-elevation area on the inside of the bar, which resembles a chute channel, and identifies this as a common feature along vegetated point bars.

R2C18: Pg 9 Ln 32: Are the observed decreases/increases in velocity significant with respect to uncertainty/error?

AC18: We have added more detail to methods addressing uncertainty in velocities.

R2C19: Figures 5-7: These graphs are not easy to read. I wonder if colour could be used in addition to line style, or results separated for density & type? Furthermore, it is unclear why lateral velocities are not reported for XS2?

AC19: The revised final manuscript will include revisions to increase the clarity of these figures, including use of color. We will also add new plot for additional scenarios (e.g., lateral velocities for XS2), some of which will be in the Supplement.

R2C20: Pg 14 Ln 9: Would be helpful to show the data for each XS for Q>10, not just XS2.

AC20: We will add figures showing results for additional scenarios (but with some in Supplement)

R2C21: Pg 14 Ln 16-17: As mentioned above, it seems the side channel to the right of the patch plays an important role in conveying discharge, particularly for higher Q values. Is this process more important than channel bend processes?

AC21: As noted above, we have added text to Discussion regarding the low-elevation / chute channel on the inside of the bend, and linking to field studies on interactions among chutes, vegetation, and morphodynamics in meandering channels.

R2C22: Pg 15 Ln 8-10: I agree that results show that the impact of vegetation increases with Q, but I do not think results show that the vegetation begins to impact on channel-bend hydraulics for Q>Q2. It seems to me that even at Q=Q2 there are significant differences in velocity distributions that may, over a long period cause significantly different channel morphology?

AC22: We have revised the text here in an effort to clarify the discharge dependence of vegetation effects on hydraulics, and to emphasize that the effects are most clear from Q2 to Q10. Below Q2, inundation of vegetation is insufficient for it to have a substantial effect. We added information on what effect is detectible given our calibration of velocity.

R2C23: Pg 15 Ln 16: I do not think the results show any evidence of 'linear' trends?

AC23: Reworded

R2C24: Pg 16 Section 4.2: It would be good to quantify the correlation between sediment and vegetation, beyond the visual observation in Figure 8. Also, these patterns demonstrate the limitation of assuming constant vegetation density across the bar as mentioned earlier.

AC24: We deleted Fig. 8; our intention in including it was to show general relationships between vegetation and sediment patches, rather than to go further in quantifying correlations. We have added additional text to the Discussion (4.2) about vegetation and sedimentation on bars, drawing from literature.

R2C25: Pg 18 Ln 21: The authors mention the presence of bars with vegetation/no vegetation. This study investigates the difference of plant type (age) but this in itself is related to channel morphology (e.g. plant succession over time) and flood discharges (e.g. destroying plants or creating new bars). It would be interesting to think about how the model could be developed to introduce different vegetation types, depending upon bar age, etc.

AC25: Future versions of the model will likely have more flexibility in terms of the vegetation characteristics that can be included. However, since the model is 2D and typically calibrated to specific conditions, it would be difficult to do all that here. We believe the Kleinhans group has been working on something similar to what you are proposing, and we have added citations to the text to better represent their work, as well as explicitly identifying directions for future modeling (end of 4.1).

Please also note the supplement to this comment:
https://www.earth-surf-dynam-discuss.net/esurf-2017-56/esurf-2017-56-AC2-supplement.zip

---

## Author Comment (AC3) · 9 Dec 2017

JM Turowski (Editor) turowski@gfz-potsdam.de

EC: Dear authors,

we have received two reviews for your paper, both of which are detailed and comprehensive. From these reviews and my own reading of the manuscript, I think that currently the major short-coming lies in the statement of the research gap. Although

you have given the objectives, it is currently unclear which open research question you trying to answer and how this fits into the existing literature. Both reviewers mention that you have overlooked relevant published papers. I suggest that you identify a research gap through a detailed literature review in the introduction and state it clearly together with the research question in the final paragraph of the introduction. The specific objectives should follow out of this research question. In the discussion and conclusion, you can pick up the question and objectives and place your new insights into the body of the existing literature. The reviewers make a large number of other points, and there are open questions on methods, results and discussion. Please take all of these points seriously when revising the paper.

All the best and looking forward to your revised manuscript, Jens Turowski

AC: Dr. Turowski, We believe we have taken most of the reviewers' suggestions into consideration in our revised manuscript. In a few instances, we have responded to the reviewers that we believe suggestions was outside the scope of our work, but we addressed the majority of the comments. Major revisions to the paper include A) reframing the introduction and motivation of the research by synthesizing what we know about vegetation and channel bends from the literature; B) clarifying details concerning methodology by adding this information to the main text or referring to the Supplement, where much of the details were already housed; C) more explicitly stating assumptions of modeling approach; and D) revising the discussion by deleting portions that bordered speculative (fine-sediment deposition and channel geometry in vegetated channels) and adding in additional insights related to ecogeomorphic feedbacks and chute channels on vegetated point bars.

We have included a revised manuscript, but have not updated figures at this time. Our understanding is we would still have additional time (4 weeks) to make these additional changes. We believe the manuscript is clearer and more focused. Thanks for your consideration.

Please also note the supplement to this comment:
https://www.earth-surf-dynam-discuss.net/esurf-2017-56/esurf-2017-56-AC3-supplement.zip
* * *
**ESurfD**

---

## Author Response (AR1)

Dear authors,

we have received two reviews for your paper, both of which are detailed and comprehensive. From these reviews and my own reading of the manuscript, I think that currently the major short-coming lies in the statement of the research gap. Although you have given the objectives, it is currently unclear which open research question you trying to answer and how this fits into the existing literature. Both reviewers mention that you have overlooked relevant published papers. I suggest that you identify a research gap through a detailed literature review in the introduction and state it clearly together with the research question in the final paragraph of the introduction. The

specific objectives should follow out of this research question. In the discussion and conclusion, you can pick up the question and objectives and place your new insights into the body of the existing literature.

The reviewers make a large number of other points, and there are open questions on methods, results and discussion. Please take all of these points seriously when revising the paper.

All the best and looking forward to your revised manuscript, Jens Turowski

[Figure]

Earth Surf. Dynam. Discuss.,
https://doi.org/10.5194/esurf-2017-56-RC1, 2017

[Figure]

General comments: I think this is an interesting study. My main concerns are that the introduction needs to include more of a literature review on what is already known about vegetation effects on flow within meander bends because many of the results presented (at least in terms of overall vegetation effects, perhaps not effects of density/vegetation stage) here are similar to previous laboratory studies. I also think that much of the discussion is highly speculative, which can be fine, but often the speculation exceeds the amount of data needed to be presented to support the suggested hypotheses.

Specific comments: Page 2, line 2: I would argue that vegetation impacts on altering the flow velocity itself (e.g. mean flow velocities, velocity profiles) as stated here have been very well studied. Flow steering, in parentheses, by vegetation has also received

attention but none of the studies that have investigated this are cited here. For example, in the discussion you review many of the laboratory studies that have investigated flow in meander bends with and without vegetation. These studies already demonstrate that vegetation can steer flow toward the outer bank, which is one of the main points of this paper. It seems like these studies should be reviewed here to highlight what is already known, and what is not known that your study is trying to address. What is this study addressing that has not been previously answered? Right now the motivation for why this work is needed is not coming through in the literature review.

Page 3, line 18: A bankfull Shields number for a gravel bed river of 0.01 would imply there is no sediment transport at bankfull flow given that the critical Shields stress is typically greater than 0.03 (Buffington and Montgomery, 1997) for these rivers. It seems somewhat unlikely that there is no transport at bankfull? In addition, cross stream and downstream shear stresses, as well as Shields stresses, are mentioned in the methods but I don't ever recall them being quantified in the results or discussion (except a map of Shields stresses in Figure 4). Why are they brought up in the methods? How did you distribute the vegetation on the bar? Did it cover the entire bar? Was it only in a certain zone where you expect vegetation to establish? The results that you obtain seem like they will be highly dependent on this chosen location and extent of the vegetation patch. For example, on Page 14, line 15: It is stated that the u and v velocities on the right side of the downstream of the vegetated bar (Figure 5) approach or equal those in the thalweg and that this is more pronounced with vegetation density. This is where the effect of vegetation patch distribution comes into play, if the vegetation patch did not extend to the channel bank then this is what one might expect. How much of this result is driven just by the lack of vegetation between the bar and the channel wall (I am assuming this is what you modeled)? Is such a complete break in vegetation likely to occur in nature?

Page 14, line 25: v values are not shown for XS2, which is near the bend apex and it is stated that the presence of vegetation did not really affect the v velocities. If the case

is being made in the discussion that vegetation will change bank scour and meander migration, doesn't this result imply that at the bend apex, although the high downstream velocity core shifts toward the left bank, the actual direction of the flow is not deflected more toward this bank with the presence of vegetation? What does this mean for bank scour at the bend apex?

Page 16, line 15-16: A low velocity region on the bar would imply lower sediment fluxes, but would not necessarily imply sediment deposition, which is the divergence of the sediment flux. Sediment deposition would only occur if the vegetation did not reduce the steering of sediment (sediment supply) into the patch itself. Given that you show that sometimes flow is steered away from the bar on the bar sides, it seems likely that the vegetation will also impact how much sediment enters the bar, and therefore whether deposition occurs.

Page 18, lines 12-27. Much of this discussion does not seem directly related to any of the results presented above, and in particular the comparisons of three bars with/without vegetation to state that there is a difference in w/d and channel narrowness is highly speculative. No w/d ratios are provided for the bars to demonstrate this. I am not clear how only three cross-sections at one study site with no variation in vegetation type (just vegetated vs. not vegetated) can be used to infer that floodplains with herbaceous vegetation may not have narrower channels than those with woody vegetation. Further, although the vegetated bar does have a deeper thalweg, it seems to often have lower elevations on the bar, which is contrary to the earlier discussion that vegetation would cause higher amounts of sediment deposition on bars.

Figure 8 and associated text: Although there are definitely locations where sand is collocated with vegetation, there are also locations where sand deposits are not located around vegetation, or that vegetation patches lack sand deposits. Can you provide more quantitative data to show that sand and vegetation are correlated such as % of sand patches within a certain distance of vegetation or something similar?

[Figure]

Technical questions: Page 1, Lines8-9: You mention alternating bars and vegetation but then discuss bend hydraulics and forces. What kind of forces are you discussing here? Alternating bars do not have to be associated with bends and it is not clear how the second half of the sentence is related to the first. The rest of the abstract seems to be geared toward a bar in a bend, which would normally be called a point bar? This comment is relevant throughout the paper where bar is used. It might be better to be more specific here about what kind of bar you mean.

Page 1, Line 11: "with and without varied vegetation parameters" is not clear here. Are you eliminating the parameters or the vegetation itself? What kind of parameters?

Page 3, line 17: I don't know if the condition of "few upstream dams" implies that flow and sediment supply are relatively unregulated. You can have just one dam upstream that can completely alter the hydrology and sediment supply downstream; it is just not the number of dams that control these parameters but how the dams are operated. Do the dams not alter the flow? Does sediment bypass the dams?

Line 9, page 8: How was $U\_m$ determined? At a cross-section upstream of the vegetation that is free from the vegetation influence?

Lines 4-7, page 9: The dense vegetation case is two orders of magnitude higher than the sparse case but both are averages on the same bar. It seems like these two averages should be the same if the average of local densities is representative of what would occur at the scale of the entire bar. Is this partly driven by the scale over which the measurements were taken, in that the 20 stems/m2 value is a local measurement and therefore likely to be higher? Is 20 stems/m2 a realistic value of stem density for an entire bar; is such an average density found in real rivers over the spatial scale of a bar?

Line 11, page 9: If you are using the flow depth based on the model run without vegetation to assign Ac, won't this skew your Ac values because the actual flow depths will likely be higher in the presence of vegetation? Also in Figure 2c, there are many

lines but only three stages of vegetation growth, and it is not possible to tell which relations were actually used in the model.

Equation (4): What grain size is used and did the grain size spatially vary in the stream, and in this calculation?

Other methods: How were the stage and nearby discharge used to calculate Q? Why is stage needed and not just a drainage area correction? How many topographic cross sections were measured in the channel, what was the spacing of the cross-sections and what was the actual point density of the DEM in the channel? No information is provided as to how water surface was measured, where it was measured and how many data points were measured for a given flow? A 18 cm RMSE for flow depth could be pretty large, depending on the flow depth magnitude. How large were water surface elevation and velocity RMSE relative to the flow depths and velocities measured in the channel? How many measured/log profile velocities were compared to the modeled velocities to obtain the RMSE? How good were the log profile fits to the measured velocities; are there large errors in what you are assuming to be measured depth-averaged velocities?

Figure 5 It would help to have the direction of the v velocity (which way is negative) noted on the figure or in the caption. There really does not seem to be any change in the v velocity in the thalweg for the Q2 flow, contrary to what is stated in the figure caption.

Page 15, line 15: Can you give an example of where dense trees do not have the maximum impact on the flow velocity as stated here? I don't remember this being discussed in the results. Also, you have modeled the drag coefficient for vegetation as being a constant with vegetation density or plant size, but studies on vegetation have shown that this coefficient can change with vegetation spacing. How might this impact your results?

Page 15, line 20: It is stated that vegetation increased the magnitude of v at the down-

stream end of the channel bend in the thalweg. In the associated figure, v either did not really change with vegetation or decreased with vegetation, implying instead that cross stream flow was not necessarily directed more toward the cutbank in this cross section. Secondary circulation should be present in all of these cross-sections and therefore, the direction of the v component of velocity will likely depend on the vertical position in the flow column. So I am not sure how much information the depth-averaged v provides in terms of the process of bank erosion? Perhaps you can comment on this.

Page 15-16, lines 30-2: What is similar or different in these studies in the outdoor lab from your study and why are there differences in the studies? The discussion on what is similar or different is somewhat vague and do not really include hypothesizes why you might see different results in your model.

Page 16, lines 10-11: It is stated that the flow velocities and shear stresses in the thalweg in the upstream cross-section are reduced with vegetation but in Figure 7, u is reduced but v is increased with vegetation and it is therefore not clear what will happen to shear stress (and sediment transport and erosion), which is not shown.

Conclusion: Please see my earlier comments above about whether vegetation will cause fine sediment deposition. Certainty this is what others have found, but I am not sure that the data you present allow you to say that deposition will occur unless you assume that the sediment input to the bar is not changed. It is not clear why cross-stream sediment transport would be reduced by the changes in flow that are mentioned, can you provide more information on this? I think the statement that "previously only attributed to bars" is not entirely true given that previous studies in meander bends have shown that vegetation can direct the flow toward the opposite bank.

Figure 7: In the caption it is stated that v decreased by becoming more negative but changing from a low negative value to a higher negative value means that the velocity actually increased because the negative sign only denotes direction. I think that you mean less negative or at least that is what the figure appears to show to me but I can't

[Figure]

really tell what part of "adjacent to the patch" you are referencing here—on the left or the right side?

Comments on supplemental information:

You alternate between u and U being velocity at a given elevation above the bed. I think you should pick one.

It is not clear to me why you used the log profile fits instead of just using the measured velocity at 0.37h. Did you not always have this measured data point because of data exclusion near the water surface? It seems like using the measured values, if possible, would lead to less uncertainties than fitting a profile and then calculating a mean value from those fits. Or do you think there are large uncertainties in a given data point, making the profile fit more reliable? How many data points were used in the velocity profile fits?

Lines 44-46: I find it slightly confusing what is being compared in this sentence. RMSE of the modeled values calculated either using the log profile or the extrapolated velocity values? Does FASTMECH assume a log profile in its calculations of mean velocity? If so could this partly explain why you obtained lower RMSE when using the log profile instead of using the extrapolated values to the water surface?

Line 64 Is this a standard error or deviation?

[Figure]

Earth Surf. Dynam. Discuss.,
https://doi.org/10.5194/esurf-2017-56-RC2, 2017

[Figure]

This is an interesting study, which examines the impact of different vegetation types and densities on flow through a channel with a vegetated bar. The topic is relevant and the work builds on a significant literature in this area. While the work seems rigorous and of good quality, there are some details of the methodology that would benefit from clarification. Furthermore, the data could be better presented to improve clarity.

Major Comments:

Representation of vegetation: The authors raise the issues regarding the use of roughness coefficients for representing vegetation. Accordingly, they adopt a much more suitable drag-based approach. However, there are still potential limitations with this approach. In particular: the parameterisation of drag coefficient, the distribution of drag elements in space and the assumption of a logarithmic profile may represent sig-

nificant limitations of the study and could receive more attention in the text (see specific comments below)

Methods: There are a number of details regarding the numerical methodology which are currently not presented, but which may have a significant impact on the results (e.g. average drag force equation, grid size & type, relative errors, approximate depths, delineation of bar).

Figures & Data: Figure 2b could be presented more clearly. Figures 5-7 could be made clearer, but also some data is referred to which is not present in these figures (higher Q values for XS1 &3).

Specific comments: Pg 5 Ln 5: Is A_S defined? Appears in supplementary data, but I'm not sure it is defined in the main text?

Pg 6 Ln 12: What was the grid size used in the simulation? Was it constant for the whole domain? Was bank (wall) shear stress included too? (i.e. cells with wall boundaries too).

Pg 6 Ln 20: In Table 1 it would be helpful see the relative magnitude of errors. Errors of 0.18m in WSE and 0.36m/s in velocity seem large, but may not be relative to the mean values? Table 1 does also not provide a comprehensive overview of the calibration. E.g. which different values were used for C_d? What was the sensitivity to this value? The two LEV values are an order of magnitude apart, were any other values in between tested? What was the rationale for picking these values? Also, the table seems to suggest that a model without any vegetation performed better than the model with vegetation?

Pg 7 Ln 11: These relaxation figures mean very little out of context. Please provide brief explanation of which variables they correspond to.

Pg 7 Ln 15: Why were you unable to maintain a curvilinear grid? This is unclear. Which nodes overlapped and why? Was the model run in Cartesian grid? Section 2.2 seems

to suggest it was curvilinear (Pg 6, Ln 5). If values were converted between grids, how was this done, i.e. interpolation methods, grid sizes etc.

Pg 8 Ln 7: Presumably the model uses an equation in terms of drag force per unit volume? It would be useful to include the exact form here.

Pg 8 Ln 10: I agree with the authors that $C_D=1$ is a common first-order approximation, and probably does an ok job for the lower section of the plants where objects are likely to be cylindrical. However, for trees, with complex foliage I would expect this assumption to be less accurate. Therefore, it might be worth reflecting on the accuracy of the model at different discharges

Figure 3: How was the vegetated bar delineated? Current vegetated extent?

Pg 8 Ln 13: If I am correct, a height-dependent value of A is used (from Figure 2). However, regardless of depth, the near-bed vegetation geometry will not change. Therefore, in terms of defining near-bed processes linked to sediment transport, I wonder what the impact is of changing $A_c$ as depth increases, given that this impact may only be significant towards the top of the flow? Above a certain height, does the effect of area on bed-processes diminish?

Pg 8 Ln 12: How does the grid resolution compare with the stem density? Are the effects of a single stem artificially 'smeared' over many stems? If so, particularly for low vegetation densities, the flow patterns may not correspond well with single, isolated large area blockages, which will have a very different impact to wide-spread small blockages.

Pg 8 Ln 13-14: The flow will typically not be logarithmic where there is vegetation present. Therefore, what errors does this assumption introduce? Are the results valid?

Pg 9 Ln 5: 20 stems per square metre seems very dense for saplings and trees? Also, for such densities, is it still valid not to consider the mass blockage effect of the vegetation?

[Figure]

Pg 9 Ln 32: Decreasing velocities in the thalweg is surprising –but seems to correspond to additional flow along a separate channel to the right of the vegetated bar? It seems this is quite an important aspect which affects other results too (e.g. flow deflection into this channel for certain vegetation conditions). This could be made clearer within the discussion which frames the problem as a simple channel bend with vegetated bar.

Pg 9 Ln 32: Are the observed decreases/increases in velocity significant with respect to uncertainty/error?

Figures 5-7: These graphs are not easy to read. I wonder if colour could be used in addition to line style, or results separated for density & type? Furthermore, it is unclear why lateral velocities are not reported for XS2?

Pg 14 Ln 9: Would be helpful to show the data for each XS for Q>10, not just XS2.

Pg 14 Ln 16-17: As mentioned above, it seems the side channel to the right of the patch plays an important role in conveying discharge, particularly for higher Q values. Is this process more important than channel bend processes?

Pg 15 Ln 8-10: I agree that results show that the impact of vegetation increases with Q, but I do not think results show that the vegetation begins to impact on channel-bend hydraulics for Q>Q2. It seems to me that even at Q=Q2 there are significant differences in velocity distributions that may, over a long period cause significantly different channel morphology?

Pg 15 Ln 16: I do not think the results show any evidence of 'linear' trends?

Pg 16 Section 4.2: It would be good to quantify the correlation between sediment and vegetation, beyond the visual observation in Figure 8. Also, these patterns demonstrate the limitation of assuming constant vegetation density across the bar as mentioned earlier.

Pg 18 Ln 21: The authors mention the presence of bars with vegetation/no vegetation. This study investigates the difference of plant type (age) but this in itself is related

to channel morphology (e.g. plant succession over time) and flood discharges (e.g. destroying plants or creating new bars). It would be interesting to think about how the model could be developed to introduce different vegetation types, depending upon bar age, etc.

[Figure]

We thank the reviewers for their insightful comments, to which we have responded in detail below. Major revisions to the paper include A) reframing the introduction and motivation of the research by synthesizing what we know about vegetation and channel bends from the literature; B) clarifying details concerning methodology by adding this information to the main text or referring to the Supplement, where much of the details were already housed; C) more explicitly stating assumptions of modeling approach; and D) revising the discussion by deleting portions that bordered speculative (fine-sediment deposition and channel geometry in vegetated channels) and adding in additional insights related to ecogeomorphic feedbacks and chute channels on vegetated point bars. We believe the manuscript is clearer and more focused. Thanks for your consideration.

**Anonymous Referee #1**

General comments: I think this is an interesting study. My main concerns are that the introduction needs to include more of a literature review on what is already known about vegetation effects on flow within meander bends because many of the results presented (at least in terms of overall vegetation effects, perhaps not effects of density/vegetation stage) here are similar to previous laboratory studies. I also think that much of the discussion is highly speculative, which can be fine, but often the speculation exceeds the amount of data needed to be presented to support the suggested hypotheses.

We have rewritten much of the introduction, including moving material about previous studies (particularly flume studies) from the discussion to the introduction, and adding new text and literature citations to better represent the state of knowledge. We have reformulated the motivation for the work (knowledge gap) as a field-scale modeling approach. See next response.

Specific comments: Page 2, line 2: I would argue that vegetation impacts on altering the flow velocity itself (e.g. mean flow velocities, velocity profiles) as stated here have been very well studied. Flow steering, in parentheses, by vegetation has also received attention but none of the studies that have investigated this are cited here. For example, in the discussion you review many of the laboratory studies that have investigated flow in meander bends with and without vegetation. These studies already demonstrate that vegetation can steer flow toward the outer bank, which is one of the main points of this paper. It seems like these studies should be reviewed here to highlight what is already known, and what is not known that your study is trying to address. What is this study addressing that has not been previously answered? Right now the motivation for why this work is needed is not coming through in the literature review.

As noted above, we have substantially revised the introduction to better represent the state of knowledge and to clarify our motivation and the knowledge gap we are filling. Significant blocks of new text are as follows;

.... "Pioneer vegetation can occur on all bar types but is most likely to survive on nonmigrating bars, such as forced alternating point bars (Wintenberger et al., 2015). Plant traits including height,

frontal area, and stem flexibility vary with elevation above the baseflow channel, influencing both the susceptibility of plants to uprooting during floods and their impact on morphodynamics (Bywater-Reyes et al., 2015, 2017b; Diehl et al., 2017a; Kui et al., 2014). Vegetation effects on hydraulics, bank erosion, and channel pattern also depend on the uniformity of vegetation distribution on bars, which can vary depending on wind versus water-based dispersal mechanisms (Van Dijk et al., 2013), and on whether plants occur individually or in patches (Manners et al. 2015).

Experimental work in flumes has shown that vegetation is vital to sustaining meandering in coarse-bedded rivers (Braudrick et al., 2009). Vegetation's effect on stabilizing banks, steering flow, and impacting morphodynamics furthermore depends on seed density and stand age. Uniform vegetation on bars has been shown, experimentally, to decrease bank erosion rates, stabilize banks, and increase sinuosity of meander bends (Van Dijk et al., 2013). Gran and Paola (2001) showed that vegetation, by increasing bank strength, generates secondary currents associated with oblique bank impingement that may be more important than helical flows generated by channel curvature. Other experiments have generally suggested vegetated bars decrease velocities over the bar and push flow toward the outer bank. For example, tests in a constructed, meandering laboratory stream with two reed species planted on a sandy point bar showed that vegetation reduced velocities over the vegetated bar, increased them in the thalweg, strengthened secondary circulation, and directed secondary flow toward the outer bank (Rominger et al., 2010). Another study in the same experimental facility, but using woody seedlings planted on the point bar, also found reduced velocities in the vegetated area of the bar, with the greatest reductions at the upstream end, and the effect varied with vegetation architecture and density (Lightbody et al., 2012). In a flume study where meandering effects were simulated in a straight channel by placing dowels representing vegetation patches in alternating locations along the edges of the flume, vegetation reduced velocity within and at the edges of the vegetation patch and increased velocities near the opposite bank (Bennett et al., 2002). Experiments in a high-curvature meandering flume, in contrast, showed that vegetation inhibited high shear-stress values from reaching the outer bank (Termini, 2016), inconsistent with studies simulating moderate sinuosity channels.

….
As the above review suggests, there have been considerable advances in laboratory and computational modelling of vegetation effects on hydraulics that complement understanding of bar and bend morphodynamics and of the reciprocal interactions between riparian vegetation and river processes (Corenblit et al., 2007; Gurnell, 2014; Osterkamp and Hupp, 2010; Schnauder and Moggridge, 2009). Challenges persist, however, in representing field-scale complexities in a modelling framework to deepen insights into the feedbacks between plants, flow, and channel morphology on vegetated point bars. Here we tackle key elements of this problem by investigating the dependence of bend hydraulics on the distribution of woody vegetation, across a range of flood magnitudes, using a two-dimensional modeling approach informed by high-resolution topography and vegetation morphology data that spatially defines vegetation drag."

Page 3, line 18: A bankfull Shields number for a gravel bed river of 0.01 would imply there is no sediment transport at bankfull flow given that the critical Shields stress is typically greater than 0.03 (Buffington and Montgomery, 1997) for these rivers. It seems somewhat unlikely that there is no transport at bankfull?
We have recalculated bankfull Shields number using field observations of bankfull discharge from a broader set of locations in the study reach. The previously reported bankfull Shields

number of 0.01 was for one specific location in our study reach, as reported in Bywater-Reyes et al. (2015, WRR). Updated calculations, from field observations at four locations, indicate bankfull Shields numbers ranging from 0.01 to 0.07. Hec-Ras solutions indicate a reach-average Shields number of 0.02 for the $Q_2$ (slightly overbank) and FaSTMECH reach-average Shields number for the $Q_2$ is 0.03. These values indicate our originally reported number of 0.01 was too low and that a value of 0.03 is more accurate. The text has been revised with the new value.

In addition, cross stream and downstream shear stresses, as well as Shields stresses, are mentioned in the methods but I don't ever recall them being quantified in the results or discussion (except a map of Shields stresses in Figure 4). Why are they brought up in the methods?
The Shields stresses are a function of velocity, so the results were very similar to those shown for velocity. We chose for that reason to show only velocity and the planview Shields map. We removed the associated Shields stress equations.

How did you distribute the vegetation on the bar? Did it cover the entire bar? Was it only in a certain zone where you expect vegetation to establish? The results that you obtain seem like they will be highly dependent on this chosen location and extent of the vegetation patch. For example, on Page 14, line 15: It is stated that the u and v velocities on the right side of the downstream of the vegetated bar (Figure 5) approach or equal those in the thalweg and that this is more pronounced with vegetation density. This is where the effect of vegetation patch distribution comes into play, if the vegetation patch did not extend to the channel bank then this is what one might expect. How much of this result is driven just by the lack of vegetation between the bar and the channel wall (I am assuming this is what you modeled)? Is such a complete break in vegetation likely to occur in nature?
The polygon (vegetated area) was chosen based on the mapped extent of vegetation (Fig. 1) on the bar of focus. This bar was the location of previous work (Bywater-Reyes et al., 2015) where vegetation densities, morphologies, and uprooting susceptibilities were determined. The results indeed may be sensitive to the delineation of this polygon. As the vegetation is represented in the model, however, drag from trees is assigned based on the density. The extent of vegetation on the bar as modeled is representative of the vegetation currently on the bar and of the strand lines of vegetation recruitment. The extreme scenarios (e.g. dense tress) may be dependent on the location of the patch, but the progression of increasing density and tree size illustrates the overall effect vegetation can have on flow steering.

Page 14, line 25: v values are not shown for XS2, which is near the bend apex and it is stated that the presence of vegetation did not really affect the v velocities. If the case is being made in the discussion that vegetation will change bank scour and meander migration, doesn't this result imply that at the bend apex, although the high downstream velocity core shifts toward the left bank, the actual direction of the flow is not deflected more toward this bank with the presence of vegetation? What does this mean for bank scour at the bend apex?
We have added figures for the additional scenarios, including v values for all cross sections, to Supplement. With respect to the Discussion (which has been revised, as described below), where we discuss our results relative to Parker et al. (2011), we note that Parker et al. (2011) is based on cross-stream gradient of streamwise velocities, not of v veocities.

Page 16, line 15-16: A low velocity region on the bar would imply lower sediment fluxes, but would not necessarily imply sediment deposition, which is the divergence of the sediment flux. Sediment deposition would only occur if the vegetation did not reduce the steering of sediment

(sediment supply) into the patch itself. Given that you show that sometimes flow is steered away from the bar on the bar sides, it seems likely that the vegetation will also impact how much sediment enters the bar, and therefore whether deposition occurs.

We have reworded to indicate that fine sedimentation could occur.

Page 18, lines 12-27. Much of this discussion does not seem directly related to any of the results presented above, and in particular the comparisons of three bars with/without vegetation to state that there is a difference in w/d and channel narrowness is highly speculative. No w/d ratios are provided for the bars to demonstrate this.
I am not clear how only three cross-sections at one study site with no variation in vegetation type (just vegetated vs. not vegetated) can be used to infer that floodplains with herbaceous vegetation may not have narrower channels than those with woody vegetation. Further, although the vegetated bar does have a deeper thalweg, it seems to often have lower elevations on the bar, which is contrary to the earlier discussion that vegetation would cause higher amounts of sediment deposition on bars.

We deleted this text and associated figure. We revisesd Fig. 1 to remove the cross section locations.

Figure 8 and associated text: Although there are definitely locations where sand is collocated with vegetation, there are also locations where sand deposits are not located around vegetation, or that vegetation patches lack sand deposits. Can you provide more quantitative data to show that sand and vegetation are correlated such as % of sand patches within a certain distance of vegetation or something similar?

We deleted Fig. 8; our intention in including it was to show general relationships between vegetation and sediment patches, rather than to go further in quantifying correlations, which we consider outside the scope of this paper. We have added additional text to the Discussion (4.2) about vegetation and sedimentation on bars, drawing from literature.

Technical questions: Page 1, Lines8-9: You mention alternating bars and vegetation but then discuss bend hydraulics and forces. What kind of forces are you discussing here? Alternating bars do not have to be associated with bends and it is not clear how the second half of the sentence is related to the first. The rest of the abstract seems to be geared toward a bar in a bend, which would normally be called a point bar? This comment is relevant throughout the paper where bar is used. It might be better to be more specific here about what kind of bar you mean.

We have changed the text to specify that we are modeling a point bar. We have removed the discussion of forces from the abstract but have retained a discussion of how the hydraulics would alter forces in the Discussion, where we can elaborate more.

Page 1, Line 11: "with and without varied vegetation parameters" is not clear here. Are you eliminating the parameters or the vegetation itself? What kind of parameters?

Reworded

Page 3, line 17: I don't know if the condition of "few upstream dams" implies that flow and sediment supply are relatively unregulated. You can have just one dam upstream that can completely alter the hydrology and sediment supply downstream; it is just not the number of dams that control these parameters but how the dams are operated. Do the dams not alter the flow? Does sediment bypass the dams?

Added clarifying text ("…flow and sediment supply are relatively unaltered by flow regulation, because the only significant dam in the contributing watershed is ~120 km upstream of the study reach, on a tributary.")

Line 9, page 8: How was U_m determined? At a cross-section upstream of the vegetation that is free from the vegetation influence?
Um is the node velocity. We have added clarifying text.

Lines 4-7, page 9: The dense vegetation case is two orders of magnitude higher than the sparse case but both are averages on the same bar. It seems like these two averages should be the same if the average of local densities is representative of what would occur at the scale of the entire bar. Is this partly driven by the scale over which the measurements were taken, in that the 20 stems/m2 value is a local measurement and therefore likely to be higher? Is 20 stems/m2 a realistic value of stem density for an entire bar; is such an average density found in real rivers over the spatial scale of a bar?
Values in the range of our dense scenario (20 stems/m2) have been reported in diverse settings; we have added references. Furthermore, our objective is to investigate end-member seedling-density cases.

Line 11, page 9: If you are using the flow depth based on the model run without vegetation to assign Ac, won't this skew your Ac values because the actual flow depths will likely be higher in the presence of vegetation? Also in Figure 2c, there are many lines but only three stages of vegetation growth, and it is not possible to tell which relations were actually used in the model.
Yes, the values would be slightly skewed. It is a limitation of the method. In the revised final manuscript, we revised Fig. 2 to more clearly show which relations were used in the model.

Equation (4): What grain size is used and did the grain size spatially vary in the stream, and in this calculation?
We used the median value from data collected over the region. Added clarifying text.

Other methods: How were the stage and nearby discharge used to calculate Q? Why is stage needed and not just a drainage area correction?
Stage is needed because water surface elevations at the downstream boundary, for specific modeled discharges, are used as a model boundary condition. We therefore needed to combine data from our measurements of stage with nearby gage measurements of Q. Added clarifying text.

How many topographic cross sections were measured in the channel, what was the spacing of the cross-sections and what was the actual point density of the DEM in the channel?
Added more details to Supplement

No information is provided as to how water surface was measured, where it was measured and how many data points were measured for a given flow? A 18 cm RMSE for flow depth could be pretty large, depending on the flow depth magnitude. How large were water surface elevation and velocity RMSE relative to the flow depths and velocities measured in the channel? How many measured/log profile velocities were compared to the modeled velocities to obtain the RMSE? How good were the log profile fits to the measured velocities; are there large errors in what you are assuming to be measured depthaveraged velocities?
We added details concerning how water surface elevations were measured (and density of observations) to the Supplement. We have added the mean measured and modeled velocities for the velocity calibration to Table 1. We note that in other studies that have used FaSTMECH, velocity calibrations have similar magnitudes of error, or higher. For example, Legleiter et al., 2011 modeled the effects of a point bar on force balance of flow with FaSTMECH for a simple channel ~60m wide with a bankfull discharge of 42.5 $m^3$/s had a RMSE $\bar{U}$ of 0.27 m/s. The mean of their $\bar{U}$ was 1.57 m/s. This is quite comparable to our $\bar{U}$ calibration. Segura and Pitlick,

2015 had RMSE-Ū of 0.14 – 0.28 m/s for reaches with very small bankfull discharges (7 – 20 m³/s). Average Ū were not reported in their text, but appear to be ~1 m/s from the figures. We provide details concerning the methods used in the Ū calibration in the Supplement. We added WSE plots to Supplement as well. Methods concerning the log profiles are in the Supplement.

Figure 5 It would help to have the direction of the v velocity (which way is negative) noted on the figure or in the caption. There really does not seem to be any change in the v velocity in the thalweg for the Q2 flow, contrary to what is stated in the figure caption.
Reworded caption.

Page 15, line 15: Can you give an example of where dense trees do not have the maximum impact on the flow velocity as stated here? I don't remember this being discussed in the results. Also, you have modeled the drag coefficient for vegetation as being a constant with vegetation density or plant size, but studies on vegetation have shown that this coefficient can change with vegetation spacing. How might this impact your results?
We reemphasized the example concerning dense young trees. Vegetation drag is an often unconstrained parameter. We have added a paragraph to the Discussion (end of 4.1) about limitations of our treatment of vegetation drag.

Page 15, line 20: It is stated that vegetation increased the magnitude of v at the down- stream end of the channel bend in the thalweg. In the associated figure, v either did not really change with vegetation or decreased with vegetation, implying instead that cross stream flow was not necessarily directed more toward the cutbank in this cross section. Secondary circulation should be present in all of these cross-sections and therefore, the direction of the v component of velocity will likely depend on the vertical position in the flow column. So I am not sure how much information the depth-averaged v provides in terms of the process of bank erosion? Perhaps you can comment on this.
u became more negative. We believe our statement "Vegetation increased the magnitude of cross-stream velocity (v) at both the up- and downstream end of the channel bend by increasing cross-stream flow toward the cutbank at the head of the bar and around the toe of the bar" is accurate. A more negative number at the downstream cross section implies more steering around the toe of the bar, as stated. The reviewer's observation regarding secondary circulation statement is valid, which is a limitation of the model.

Page 15-16, lines 30-2: What is similar or different in these studies in the outdoor lab from your study and why are there differences in the studies? The discussion on what is similar or different is somewhat vague and do not really include hypothesizes why you might see different results in your model.
Moved info to introduction and placed study within the context of what is known.

Page 16, lines 10-11: It is stated that the flow velocities and shear stresses in the thalweg in the upstream cross-section are reduced with vegetation but in Figure 7, u is reduced but v is increased with vegetation and it is therefore not clear what will happen to shear stress (and sediment transport and erosion), which is not shown.
Figure 4 shows the Shields number, which is reduced

Conclusion: Please see my earlier comments above about whether vegetation will cause fine sediment deposition. Certainty this is what others have found, but I am not sure that the data you present allow you to say that deposition will occur unless you assume that the sediment input to the bar is not changed. It is not clear why cross-stream sediment transport would be

reduced by the changes in flow that are mentioned, can you provide more information on this? I think the statement that "previously only attributed to bars" is not entirely true given that previous studies in meander bends have shown that vegetation can direct the flow toward the opposite bank.

We have rephrased

Figure 7: In the caption it is stated that v decreased by becoming more negative but changing from a low negative value to a higher negative value means that the velocity actually increased because the negative sign only denotes direction. I think that you mean less negative or at least that is what the figure appears to show to me but I can't really tell what part of "adjacent to the patch" you are referencing hereâ˘A ˘Ton the left or the right side?

Reworded

Comments on supplemental information:
You alternate between u and U being velocity at a given elevation above the bed. I think you should pick one.

Fixed

It is not clear to me why you used the log profile fits instead of just using the measured velocity at 0.37h. Did you not always have this measured data point because of data exclusion near the water surface? It seems like using the measured values, if possible, would lead to less uncertainties than fitting a profile and then calculating a mean value from those fits. Or do you think there are large uncertainties in a given data point, making the profile fit more reliable? How many data points were used in the velocity profile fits?

We have missing values. Profiles were fit with a minimum of four points.

Lines 44-46: I find it slightly confusing what is being compared in this sentence. RMSE of the modeled values calculated either using the log profile or the extrapolated velocity values? Does FASTMECH assume a log profile in its calculations of mean velocity? If so could this partly explain why you obtained lower RMSE when using the log profile instead of using the extrapolated values to the water surface?

This is certainly possible, but since the values are missing, we can't really know the real difference. Yes, the model assumes a logarithmic profile. We followed a published procedure.

Line 64 Is this a standard error or deviation?

Added "standard deviation"

This is an interesting study, which examines the impact of different vegetation types and densities on flow through a channel with a vegetated bar. The topic is relevant and the work builds on a significant literature in this area. While the work seems rigorous and of good quality, there are some details of the methodology that would benefit from clarification. Furthermore, the data could be better presented to improve clarity.

We have clarified methodology questions and have improved figures, in response to specific reviewer suggestions.

Major Comments:
Representation of vegetation: The authors raise the issues regarding the use of roughness coefficients for representing vegetation. Accordingly, they adopt a much more suitable drag-based approach. However, there are still potential limitations with this approach. In particular: the parameterisation of drag coefficient, the distribution of drag elements in space and the assumption of a logarithmic profile may represent significant limitations of the study and could receive more attention in the text (see specific comments below)

We recognize that our modeling does not fully represent the complexities of field-based vegetation and flow conditions; we have added or revised text in several locations to highlight our assumptions and/or limitations, including a paragraph in the Discussion (end of 4.1) explicitly discussing these issues.

Methods: There are a number of details regarding the numerical methodology which are currently not presented, but which may have a significant impact on the results (e.g. average drag force equation, grid size & type, relative errors, approximate depths, delineation of bar).

Both reviewers requested clarifications on modeling details. We have added text to address these details in both the main text and the Supplement (in some cases, the information requested by reviewers was in the Supplement in the original version). We have added a sentence pointing readers to the Supplement early in Methods. In some cases, we have moved details that were previously in the Supplement to the main text, in response to review comments, but some details we consider more suitable for the Supplement.

Figures & Data: Figure 2b could be presented more clearly. Figures 5-7 could be made clearer, but also some data is referred to which is not present in these figures (higher Q values for XS1 &3).

The revised manuscript includes revisions to increase the clarity of several figures. Specifically, we use the average curve for each seedling size to Figure 2. For Figures 5-7 we added colors and include all figure combinations; important examples will be in the main text and additional combinations, of which there are many, are presented in the Supplement.

Specific comments: Pg 5 Ln 5: Is A_S defined? Appears in supplementary data, but I'm not sure it is defined in the main text?

Changed to $A_c$ to reduce confusion/simplify

Pg 6 Ln 12: What was the grid size used in the simulation? Was it constant for the whole domain? Was bank (wall) shear stress included too? (i.e. cells with wall boundaries too).
We have added details on grid resolution to the main text (2.5 x 2.5 m cells for calibration runs, 5 x 5 m cells for remaining runs); they are also in the Supplement. The grid size was constant for the whole domain (although as noted, differed between calibration and other runs). We added a reminder in the main text to point readers to the Supplement. Wall stresses were not calculated. These should be negligible in the channel modeled here, where width >> depth.

Pg 6 Ln 20: In Table 1 it would be helpful see the relative magnitude of errors. Errors of 0.18m in WSE and 0.36m/s in velocity seem large, but may not be relative to the mean values? Table 1 does also not provide a comprehensive overview of the calibration. E.g. which different values were used for C_d? What was the sensitivity to this value? The two LEV values are an order of magnitude apart, were any other values in between tested? What was the rationale for picking these values? Also, the table seems to suggest that a model without any vegetation performed better than the model with vegetation?
We added more detail to Table 1 and the Supplement concerning WSE and Ū calibration. We added details to the text concerning the range of LEV and Cd values tested. The model with vegetation for $Q_2$ (453 m3/s) did perform slightly better in terms of WSE, but by a minimal amount. We do not have Ū measurements for this flow.

Pg 7 Ln 11: These relaxation figures mean very little out of context. Please provide brief explanation of which variables they correspond to.
Added clarifying text ("FaSTMECH uses relaxation coefficients to control changes in a parameter between iterations (Nelson, 2013). Relaxation coefficients were set to 0.5, 0.3, and 0.1 for ERelax, URelax, and ARelax, respectively, through trial and error.")

Pg 7 Ln 15: Why were you unable to maintain a curvilinear grid? This is unclear. Which nodes overlapped and why? Was the model run in Cartesian grid? Section 2.2 seems to suggest it was curvilinear (Pg 6, Ln 5). If values were converted between grids, how was this done, i.e. interpolation methods, grid sizes etc.
We added the text "We were unable to maintain a curvilinear, channel-fitted grid (nodes overlapped) so we projected our Cartesian coordinate flow solution output to the nearest grid cell of a curvilinear grid (2 by 2 average grid resolution) covering the main channel, and converted the associated output to streamwise and stream-normal values with a rotation matrix. A piecewise Cubic Hermite Interpolating Polynomial algorithm was applied to reduce artifacts from the transformation" to the Supplement.

Pg 8 Ln 7: Presumably the model uses an equation in terms of drag force per unit volume? It would be useful to include the exact form here.
Added detail that drag is averaged over vegetation polygons. Because the model is 2D, drag force is per bed area, not volume.

Pg 8 Ln 10: I agree with the authors that C_D=1 is a common first-order approximation, and probably does an ok job for the lower section of the plants where objects are likely to be cylindrical. However, for trees, with complex foliage I would expect this assumption to be less accurate. Therefore, it might be worth reflecting on the accuracy of the model at different discharges
We added a paragraph in the Discussion (end of 4.1) discussing these issues.

Figure 3: How was the vegetated bar delineated? Current vegetated extent?

Vegetated bar was delineated based on current mapped vegetation extent (Fig. 1), as indicated in the main text and the Supplement. (one of our responses to Referee 1 also addresses delineation of the vegetated bar)

Pg 8 Ln 13: If I am correct, a height-dependent value of A is used (from Figure 2). However, regardless of depth, the near-bed vegetation geometry will not change. Therefore, in terms of defining near-bed processes linked to sediment transport, I wonder what the impact is of changing A_c as depth increases, given that this impact may only be significant towards the top of the flow? Above a certain height, does the effect of area on bed-processes diminish?
This is correct, a height-dependent value of frontal area is used, from Fig. 2 (which we revised for clarity; we have also made minor revisions to the caption for clarity). As shown in Fig. 2, the height dependence of $A_c$ is most important for depths between ~0.2 and 1 m, with variations among growth stages, and diminishing effects at greater heights (Fig. 2). We agree that near-bed processes most linked to sediment transport are not fully captured by this approach. Given our focus on hydraulics, rather than near-bed sediment transport processes, we consider our approach to be adequate. Indeed, we consider using field measurements of vegetation structure with ground-based LiDAR to determine frontal area and variations with height, for different growth stages, and incorporation of height / depth-dependence of frontal area into modeling, to be an advance over standard modeling practices and a strength of our study.

Pg 8 Ln 12: How does the grid resolution compare with the stem density? Are the effects of a single stem artificially 'smeared' over many stems? If so, particularly for low vegetation densities, the flow patterns may not correspond well with single, isolated large area blockages, which will have a very different impact to wide-spread small blockages.
We have added details on grid resolution to the main text; they are also in the Supplement. Stem density is used to calculate projected vertical frontal area of vegetation and vegetation form drag (eq. 1). Our intent here is not to represent the effects of vegetation at all scales, but rather to assess two end-member density and vegetation drag scenarios. We recognize the complexity of vegetation affecting hydraulics at multiple scales as a function of patch configuration. We have treated these topics in other papers (see response to comment below). We reference Vargas-Luna et al. (2015a) in that representing vegetation as cylinders averaged over an area works best for dense vegetation.

Pg 8 Ln 13-14: The flow will typically not be logarithmic where there is vegetation present. Therefore, what errors does this assumption introduce? Are the results valid?
We recognize that vegetation will disrupt logarithmic velocity profiles, and we agree that a complete representation of vegetation effects on the velocity profile is a worthy goal, albeit one that we consider beyond our scope. We added text more explicitly recognizing the limitation of assuming a log velocity profile ("The model assumes a logarithmic velocity profile, although we recognize this is an over-simplification of how factors such as vegetation submergence alter velocity profiles (e.g., Manners et al., 2015)." In general (including via revisions in response to comments here) we have sought to be transparent about the limitations of our modeling approach, and to emphasize results and insights that we consider valid even in light of those limitations.

Pg 9 Ln 5: 20 stems per square metre seems very dense for saplings and trees? Also, for such densities, is it still valid not to consider the mass blockage effect of the vegetation?
Densities of 20 stems / m2 are indeed dense, but are consistent with literature values; we have added references. Furthermore, our objective is to investigate end-member cases.

With respect to the second part of the comment, regarding mass blockage effect, we agree that this could be an important effect for larger-diameter plants. For the size (diameter) of plants in

our field site, even at the high densities considered here, we do not expect plants to act as collective bodies with mass blockage effects. We have thought extensively about the relationship between vegetation morphology and organization on hydraulics. In Bywater-Reyes et al. (2017, JGR-ES), we use terrestrial laser scans of woody seedlings to measure roughness density, blockage effects, and implications for hydraulic structures. In Manners et al. (2015, JGR-ES) and Diehl et al. (2017, ESPL), we measure (in a flume) how woody seedlings differentially affect hydraulics and topography depending on whether they are organized individually or in patches.

Pg 9 Ln 32: Decreasing velocities in the thalweg is surprising –but seems to correspond to additional flow along a separate channel to the right of the vegetated bar? It seems this is quite an important aspect which affects other results too (e.g. flow deflection into this channel for certain vegetation conditions). This could be made clearer within the discussion which frames the problem as a simple channel bend with vegetated bar.
We have added text to the Discussion (section 4.2) about the low-elevation area on the inside of the bar, which resembles a chute channel, and identifies this as a common feature along vegetated point bars.

Pg 9 Ln 32: Are the observed decreases/increases in velocity significant with respect to uncertainty/error?
We have added more detail to methods addressing uncertainty in velocities.

Figures 5-7: These graphs are not easy to read. I wonder if colour could be used in addition to line style, or results separated for density & type? Furthermore, it is unclear why lateral velocities are not reported for XS2?
The revised final manuscript includes revisions to increase the clarity of these figures, including use of color. We also added new plot for additional scenarios (e.g., lateral velocities for XS2), in the Supplement.

Pg 14 Ln 9: Would be helpful to show the data for each XS for Q>10, not just XS2.
We added figures showing results for additional scenarios (in Supplement)

Pg 14 Ln 16-17: As mentioned above, it seems the side channel to the right of the patch plays an important role in conveying discharge, particularly for higher Q values. Is this process more important than channel bend processes?
As noted above, we have added text to Discussion regarding the low-elevation / chute channel on the inside of the bend, and linking to field studies on interactions among chutes, vegetation, and morphodynamics in meandering channels.

Pg 15 Ln 8-10: I agree that results show that the impact of vegetation increases with Q, but I do not think results show that the vegetation begins to impact on channel-bend hydraulics for Q>Q2. It seems to me that even at Q=Q2 there are significant differences in velocity distributions that may, over a long period cause significantly different channel morphology?
We have revised the text here in an effort to clarify the discharge dependence of vegetation effects on hydraulics, and to emphasize that the effects are most clear from Q2 to Q10. Below Q2, inundation of vegetation is insufficient for it to have a substantial effect. We added information on what effect is detectible given our calibration of velocity.

Pg 15 Ln 16: I do not think the results show any evidence of 'linear' trends?

Reworded

Pg 16 Section 4.2: It would be good to quantify the correlation between sediment and vegetation, beyond the visual observation in Figure 8. Also, these patterns demonstrate the limitation of assuming constant vegetation density across the bar as mentioned earlier.
We deleted Fig. 8; our intention in including it was to show general relationships between vegetation and sediment patches, rather than to go further in quantifying correlations. We have added additional text to the Discussion (4.2) about vegetation and sedimentation on bars, drawing from literature.

Pg 18 Ln 21: The authors mention the presence of bars with vegetation/no vegetation.
This study investigates the difference of plant type (age) but this in itself is related to channel morphology (e.g. plant succession over time) and flood discharges (e.g. destroying plants or creating new bars). It would be interesting to think about how the model could be developed to introduce different vegetation types, depending upon bar age, etc.
Future versions of the model will likely have more flexibility in terms of the vegetation characteristics that can be included. However, since the model is 2D and typically calibrated to specific conditions, it would be difficult to do all that here. We believe the Kleinhans group has been working on something similar to what you are proposing, and we have added citations to the text to better represent their work, as well as explicitly identifying directions for future modeling (end of 4.1).

[revised manuscript text omitted]

---

## Author Response (AR2)

**EC:**

Associate Editor Decision: Reconsider after major revisions (26 Jan 2018) by Jens Turowski

Comments to the Author:

Dear authors,

I was lucky to be able to solicit the two original reviewers again. While one of the was entirely happy with your edits, the other has some further questions. These mainly relate to details of the methods and accuracy of the modelling. I expect that the amount of revisions needed to bring the manuscript to a publishable state are moderate. I will likely send out the revised paper to the critical reviewer again for screening, but I don't think another round of full reviews will be necessary (depending on your edits and rebuttal, of course).

Thanks for your efforts and I am looking forward to seeing your revised paper.

All the best, Jens Turowski

**AC:**
**Dr. Turowski, we thank your team and reviewers for their continued diligent work reviewing this manuscript. We have addressed all comments to the best of our ability. The majority of the changes were in clarifying remaining issues raised by R1 and R2, which were mostly editorial in nature. R1C2 was the exception, where R1 raises skepticism concerning the accuracy of our model results that consider vegetation drag. Skepticism is certainly warranted. We cite the literature that has shown accounting for vegetation drag in a manner similar to our approach is more accurate and appropriate than traditional roughness approaches, starting p. 3, L14 of the introduction. However, our objective is not to illustrate improved accuracy with the use of the model. Rather, our study applies a framework, tested by others, to a field setting to understand potential interactions between vegetation, flow, and channel-bend flow dynamics (p. 3, L30). Such field studies are sparse. As such, the accuracy of our model will need to be tested in additional studies. Our results provide a framework to do so by making testable predictions about how rivers with bare vs. vegetated bars behave. We believe this work is valuable in spite of its caveats, which we fully acknowledge.**

Sincerely,

Sharon, Rebecca, and Andrew

Report #1

The authors have responded clearly to my comments and have improved the manuscript. However, I still have a
few concerns regarding the methods used.

**R1C1:**
1. Unfortunately I still cannot work out the precise meaning of the text in the supplementary material, describing the gridding process. This may seem like a minor point, and I apologise if I am missing something, but I think it is important as others may wish to reproduce the modelling methods. It appears from the text that a curvilinear grid was created from the topography (2.5m); the authors then ran the model in Cartesian co-ordinates (topography mapped from the previous grid?) and then mapped these Cartesian velocities back on to a different curvilinear grid with higher resolution (2m)? The text suggests Cartesian velocities (Pg 8,Ln 17. Figs 3 &4) but the text also mentions that FASTMECH solves in streamwise and cross-stream co-ordinates (Pg 7, Ln 4-5)? Therefore, could the authors clarify:
• Was the hydraulic model run in Cartesian or orthogonal curvilinear co-ordinates? Why were two grid conversions
required?
• What is meant by the "average cell size" (Supp pg 1, Ln 13) if "the grid size was constant for the whole domain"
(Supp pg 1, Ln 15)? Does it mean constant area but not constant length/width?
• What precisely is meant by "unable to maintain a curvilinear, channel-fitted grid (nodes overlapped)"
**AC1:**
**FASTMECH indeed does solve streamwise and cross-stream coordinates on an orthogonal curvilinear grid. To create the grid, the user picks a centerline, the width of the grid, and then can change the number of i's and j's. Here is a screenshot of what this looks like.**

[Figure]

**The solver manual (and the developers, personal communication) suggest not picking the exact centerline of the channel, as this is not practical. We provide an example below that is extreme but illustrative:**

[Figure]

**Because a sharp bend is present, the nodes overlap and the grid cannot be created.**

**Instead of following the exact center of the channel, it is recommended that one picks the "centerline" in more practical terms, such that on average, the grid and model domain is centered over the area of interest. Here is our model domain:**

[Figure]

The center of our grid is more or less over the channel, but cannot follow the center of the channel exactly or the nodes will overlap. Thus, the model was run on an orthogonal curvilinear grid, but not one that follows the channel bend of interest. Thus, the results (relative to a Cartesian grid; an output option in FaSTMECH) were rotated to one fitting the channel bend of interest. So only one conversion was done.

The image of our grid shows that the cells are not exactly the same size, thus the "average" value. The terminology "The grid size was constant for the whole domain" is perhaps misleading. We meant that we did not "change resolution" anywhere along the centerline, but this was not clear. We deleted this sentence.

The grid on which the solution output was projected was higher resolution (2 by 2 m) simply so the grid of solutions were not overly averaged by projecting onto a coarse grid.

We have made some revisions to the supplement. If this is still unclear (here or in the manuscript/supplement) please let us know how we might make this clearer.

**R1C2:**
2. In response to my initial comments, the authors confirm that for the comparison case in the calibration, the unvegetated model outperforms the vegetated model. This is not necessarily surprising if the un-vegetated model has been pre-calibrated using $C_d$ (i.e. vegetation possibly already accounted for in bed $C_d$). However, it does question the accuracy and usefulness of the vegetation model component and the resulting analysis. I agree that even if the vegetation model shows a higher RMSE in WSE, it might perform better at predicting local velocities, but I cannot see any evidence of this presented here. The results show that the vegetation model produces different results, but not necessarily more accurate results. Is it possible to use the field data to show that the vegetation model enhances prediction of local velocity patterns (i.e. apply the vegetation model at a different discharge where velocity data are available for calibration/validation)? At present, although I follow the logic of model development, based on the evidence presented I find the case for the application of the vegetation model unconvincing
**AC2:**
**We agree that this model imperfectly represents field / flood / vegetation conditions, and that skepticism is appropriate. The velocity calibration data, for example, were only collected during a low-flow condition, not during vegetation inundation, when collection of velocity data would have been challenging from a logistics and safety perspective. We maintain that the results of the study are still useful in making predictions about how a channel with a vegetated bar may behave differently than an unvegetated one, but the model will need more testing. We have defined future research objectives p. 20, L13. Please also see EC response.**

**R1C3:**
3. It would be helpful to include the drag equation in its area-averaged form rather than in the single stem form given, so that the stem density, which is listed as a key variable, appears explicitly, as it would have done in the actual model equation. I expect it would look something like that in Nepf (1999, as already cited in the text) although that paper states it per unit mass.
**AC3:**
**We added stem density to the drag equation in order to represent the true drag force per unit area (i.e., stem area * # of stems in a square meter).**

**R1C4:**
4. In Figure 5 what is the j co-ordinate unit, cells? and is j defined as the cross-stream co-ordinate in the text?
Might units of metres be better for comparing with other figures?
**AC4:**
**The j co-ordinates were defined in one of the iterations... Looks like it accidently got removed! We have changed this to distance in the figures and thus removed the j coordinate notation.**

**R1C5:**

5. Pg 20 Line 11: typo: "…to be most…"
**AC5:**
**Done**

Report #2

I have read both versions of the submitted manuscript and think that the authors have addressed all of my previous major concerns. I only have minor editorial comments to help with clarification of some points.

**R2C1:**
Page 3, lines 14-29: I am a little confused about the topic of this paragraph, it begins with discussing how simply adding vegetation resistance corrections can be problematic and that use of cylinder drag may be better. It is clear that Vargas-Luna used cylinders, and presumably Iwasaki et al. did with drag corrections? What about Marjoribanks et al.? Can you clarify?
**AC1:**
**We have clarified these were modeled as cylinders for all studies.**

**R2C2:**
Page 4, line 20: do you mean drainage area "is"
**AC2:**
**Changed**

**R2C3:**
Page 8, line 8, extra period here.
**AC3:**
**Changed**

**R2C4:**
Page 19, line 1: I still think there is some confusion about negative v meaning a decrease in velocity that is carried over from the previous version of the manuscript. In the Figure 7 v becomes more negative to the right of the patch and you state this in the text but in parenthesis, you state that v was reduced by 180%. I don't think this is
true, if it has become more negative, it must have increased? Or am I misunderstanding something? If this is true, can you please make sure to go through all of the cases in which v becomes more negative and make sure that you do not say that it decreased?
**AC4:**
**Thank you for noticing. We have changed throughout.**

**R2C5:**
Page 19, line 9 "Laboratory studies by Blanckaert (2010), representing sharp meander bends, identified relationships between zones of inward versus outward mass transport, transverse bed profiles, and curvature variations" I feel like this statement is pretty general and doesn't actually describe the relations between all of these parameters, which makes it difficult to understand what the main point is here as it relates to your paper and the surrounding sentences. Maybe you could clarify some?
**AC5:**
**The text has been revised to address this comment and more closely fit with rest of our paper; new text is: "Laboratory studies by Blanckaert (2010), representing sharp meander bends, illustrated that curvature-induced secondary flow associated with topographic steering concentrates most discharge over the deepest, outer parts of a bend and influences bed topography via vertical, downwelling velocities that contribute to pool scour and inward, near-bed velocities that help maintain steep, transverse bed slopes."**

**R2C6:**
Page 20, first paragraph: This seems to be more of a restatement of the results than a discussion of what the results mean, why the results occurred, or how they relate to the previous literature. I am not really clear about the main goal of this paragraph?
**AC6:**
**The main goal of this paragraph was to link our model results to broader work regarding flow steering across meander bends. Upon consideration of both this comment and the previous one (regarding Blanckaert's work), we have reorganized the text so that our**

**model results regarding flow steering follow directly on the literature discussion (Dietrich and Smith, and Blanckaert) regarding flow steering; this is now the first paragraph of section 4.1.**

**R2C7:**
Page 21, line 7 "Our analysis, more comparable to the flood-dispersed case, shows the potential for development of vegetated islands, but also for prevention of chute cutoff through bar-head maintenance; chute cutoff may be more likely in the absence of vegetation (Constantine et al., 2010)." But your analyses also show that the velocities within a potential chute on the point bar are increased by the presence of vegetation. So the velocity may be lower at the bar head with vegetation, but if velocities are increasing downstream in the incipient chute, couldn't you have upstream migration of erosion within the chute that counteracts any potential deposition at the bar head? Just a thought….

**AC71:**
**This is an excellent point: as the reviewer recognizes, the literature on chute cutoffs proposes mechanisms whereby cutoffs are initiated at the upstream end of the bar, or by upstream migration within the chute. We have slightly nuanced our text by changing "prevention" to "obstruction," although we refrain from delving into a deeper discussion of chute cutoff mechanisms here—this is certainly an interesting topic, but one which we consider beyond our scope, and one to which our modeling cannot substantively contribute (although we consider the text here, which was added in response to previous review comments, to be defensible).**

**R2C8:**
Page 21, line 22. Does HIPS abbreviate something? Please define.

**AC8:**
**We have revised the sentence to remove "HIPS" while retaining key content.**

**R2C9:**
Page 21, line 28 Can you please briefly explain this "from increasing bank erosion rates that would tend to increase sinuosity," Why would higher bank erosion rates result in a higher sinuosity?

**AC9:**
**We agree this sentence was confusing. We have deleted it entirely.**

[revised manuscript text omitted]
, as is appropriate in rivers. RTK point density was 1.25 pts m$^{-2}$. All topographic points were combined in iRIC, from which we made and a a curvilinear orthogonal grid with a centerline following the general pattern of the channel over the model domain created with an average cell size of 2.5 by 2.5 m for calibration runs, and 5 by 5 m for the remaining runs, with corresponding 841,851 and 210,926 nodes, respectively. The grid size was constant for the whole domain. We were unable to maintain a curvilinear, channel-fitted grid (nodes overlapped) so wWe projected our Cartesian coordinate flow solution output to the nearest grid cell of a curvilinear grid (2 by 2 m average grid resolution) covering the channel bend of interest (Figure 3)main channel, and converted the associated output to streamwise and stream-normal values with a rotation matrix. A piecewise Cubic Hermite Interpolating Polynomial algorithm was applied to reduce artifacts from the transformation.

**Model calibration**

We surveyed water surface elevation (WSE) with RTK GPS in at least 30 WSE locations per calibration over a 180 m reach length for each calibration flow (see main text). The calibrated runs (Table 1; Fig. S1) had RMSE of 0.11 – to 0.18 m.

[Figure]

**Figure S1. Water surface elevation (WSE) calibration for runs run 1 (a), 2 (b), 3 (c), 4 and 5 (d) (Table 1).**

[revised manuscript text omitted]

Average polygon attributes were calculated (vegetation density (#stems m$^{-2}$), height (m), diameter (m), and $A_C$ (average flow depth multiplied by average diameter at breast height; m$^2$

per plant).

**Supplemental results figures**

[Figure]

[Figure]

**Figure S2. Effect of the vegetated bar (j = 33–75) on the streamwise (u; a,b) and stream-normal (v; c,d) velocity at the**
**downstream cross section (XS1) for the Q$_{20}$ (a,c) and Q$_{100}$ (b,d) flows, with distance from river right end point (Figure 4).**
**With increasing discharge, plant size (seedling to young trees) and density, u is increased and v decreased within the**
**thalweg (j = 100). Both u and v (positive downstream and toward left bank, respectively) are decreased over the bar, and**
**for the sparse young trees and all dense scenarios increased at the edge of the patch. The results for the Q$_{20}$ and Q$_{100}$,**
**shown here, are similar to the Q$_{10}$ results (Figure 5).**

[Figure]

[Figure]

**Figure S3. Effect of the vegetated bar (j = 32–82) on the stream-normal (*v*) velocity at the midstream cross section (XS2)**
**for the Q$_2$ (a), Q$_{10}$ (b), Q$_{20}$ (c), and Q$_{100}$ (d) flows, with distance from river right end point (Figure 4). In general, *v* values**
**are much smaller than *u* values at XS2 (see Figure 6), and not substantially influenced by bar vegetation.**

[Figure]

[Figure]

Figure S4. Effect of the vegetated bar (j = 50–65) on the streamwise (*u*; a,b) and stream-normal (*v*; c,d) velocity at the upstream cross section (XS1) for the $Q_{20}$ (a,c) and $Q_{100}$ (b,d) flows, with distance from river right end point (Figure 4). In the thalweg (j = 90) and at the head of the bar, *u* is decreased with increasing seedling size and density. For $Q \geq Q_{10}$, *v* became more negative adjacent to the vegetation patch. The results for the $Q_{20}$ and $Q_{100}$ are similar to that of the $Q_{10}$ flow (shown in Figure 7).

**List of terms**

$A_c$ = vegetation frontal area (m$^2$)
$c_{Uz}$ = intercept from regression of $U$ as a function of $z$

$m_{Uz}$ = slope of regression of $U$ as a function $z$

$u_*$ = shear velocity

$u_{*_{Uz}}$ = shear velocity calculated from regression $U$ as a function of $z$

$\bar{U}$ = depth-averaged velocity (m s$^{-1}$)

$U$ = velocity (m s$^{-1}$)

$z_m$ = height above bed corresponding to law-of-wall-predicted average velocity

$z_o$ = roughness height (m)

$z_{o_{Uz}}$ = roughness height (m) determined from regressing $U$ as a function of $z$

$\kappa$ = von Karman constant

**References cited**

Bergeron, N. and Abrahams, A.: Estimating shear velocity and roughness length from velocit
profiles, Water Resour. Res., 28(8), 2155–2158, 1992.

Hemery, G. E., Savill, P. S. and Pryor, S. N.: Applications of the crown diameter–stem diameter
relationship for different species of broadleaved trees, For. Ecol. Manage., 215(1–3), 285–294,
doi:10.1016/j.foreco.2005.05.016, 2005.

Koch, B., Heyder, U. and Weinacker, H.: Detection of Individual Tree Crowns in Airborne Lidar
Data, Photogramm. Eng. Remote Sens., 72(4), 357–363, doi:10.14358/PERS.72.4.357, 2006.

Rennie, C. D. and Church, M.: Mapping spatial distributions and uncertainty of water and
sediment flux in a large gravel bed river reach using an acoustic Doppler current profiler, J.
Geophys. Res., 115(F3), doi:10.1029/2009JF001556, 2010.

Rennie, C. D. and Millar, R. G.: Measurement of the spatial distribution of fluvial bedload
transport velocity in both sand and gravel, Earth Surf. Process. Landforms, 29(10), 1173–1193,
doi:10.1002/esp.1074, 2004.

Venditti, J. G., Domarad, N., Church, M. and Rennie, C. D.: The gravel-sand transition:
Sediment dynamics in a diffuse extension, J. Geophys. Res. Earth Surf., 120, 1–21, doi:10.1002/2014JF003328.Received, 2015.